# Interpreting population- and family-based genome-wide association studies in the presence of confounding

**Carl Veller** [1] *, **Graham M. Coop** [2] *

1 Department of Ecology & Evolution, University of Chicago, Chicago, Illinois, United States of America,
2 Department of Evolution and Ecology, and Center for Population Biology, University of California, Davis, California, United States of America

* carl.veller@gmail.com (CV); gmcoop@ucdavis.edu (GMC)

## Abstract

A central aim of genome-wide association studies (GWASs) is to estimate direct genetic effects: the causal effects on an individual's phenotype of the alleles that they carry. However, estimates of direct effects can be subject to genetic and environmental confounding and can also absorb the "indirect" genetic effects of relatives' genotypes. Recently, an important development in controlling for these confounds has been the use of within-family GWASs, which, because of the randomness of mendelian segregation within pedigrees, are often interpreted as producing unbiased estimates of direct effects. Here, we present a general theoretical analysis of the influence of confounding in standard population-based and within-family GWASs. We show that, contrary to common interpretation, family-based estimates of direct effects can be biased by genetic confounding. In humans, such biases will often be small per-locus, but can be compounded when effect-size estimates are used in polygenic scores (PGSs). We illustrate the influence of genetic confounding on population- and family-based estimates of direct effects using models of assortative mating, population stratification, and stabilizing selection on GWAS traits. We further show how family-based estimates of indirect genetic effects, based on comparisons of parentally transmitted and untransmitted alleles, can suffer substantial genetic confounding. We conclude that, while family-based studies have placed GWAS estimation on a more rigorous footing, they carry subtle issues of interpretation that arise from confounding.

## 1 Introduction

Genome-wide association studies (GWASs) have identified thousands of genetic variants that are associated with a wide variety of traits in humans. In the standard "population-based" approach, the GWAS is conducted on a set of "unrelated" individuals. The associations that are detected can arise when a variant causally affects the trait or when it is in tight physical linkage with causal variants nearby.

Central to the aims of GWASs is the estimation of variants' effect sizes on traits of interest. These effect-size estimates are important for identifying and prioritizing variants and

10520811 and https://github.com/cveller/confoundedGWAS.

**Funding:** Funding was provided by the National Institutes of Health (NIH R35 GM136290 awarded to GC) and a Branco Weiss fellowship to CV. The funders had no role in study design, data collection and analysis, decision to publish, or preparation of the manuscript.

**Competing interests:** The authors have declared that no competing interests exist.

implicated genes for functional followup, and may be used to form statistical predictors of trait values or to understand the causal or mechanistic role of genetic variation in traits. Understanding sources of error and bias in GWAS effect-size estimates is therefore crucial.

The interpretation of GWAS effect-size estimates is complicated by 4 broad factors [1,2]. First, the causal pathways from an allele to phenotypic variation need not reside in the individuals who enrolled in the GWAS, but can also reflect causal effects on the individual's environment of the genotypes of their siblings, parents, other ancestors, and neighbors (indirect genetic effects or dynastic effects) [3]. Second, a phenotypic association can result from correlations between the allele and environmental causes of trait variation (environmental confounding) [4]. Third, a phenotypic association can be generated at a locus if it is genetically correlated with causal loci outside of its immediate genomic region (genetic confounding) [1]. Fourth, an allele's effect on a trait might depend on the environment and the allele's genetic background (gene–environment and gene–gene interactions, or G×E and G×G) [5–7].

Since our primary interest here will be genetic confounding, we briefly describe some potential sources of the long-range allelic associations that drive it: population structure, assortative mating, and selection on the GWAS trait.

Population structure leads to genetic correlations across the genome when allele frequencies differ across populations or geographic regions: sampled individuals from particular populations are likely to carry, across their genomes, alleles that are common in those populations, which induces correlations among these alleles, potentially across large genomic distances. Such genetic correlations persist even after the populations mix, as alleles that were more common in a particular source population retain their association until uncoupled by recombination.

Assortative mating brings alleles with the same directional effect on a trait (or on multiple traits, in the case of cross-trait assortative mating) together in mates, and therefore, bundles these alleles in offspring and subsequent generations. This bundling manifests as positive genetic correlations among alleles with the same directional effect [8,9], which can confound effect-size estimates in a GWAS on the trait.

Finally, natural selection on a GWAS trait can result in genetic correlations by favoring certain combinations of trait-increasing and trait-decreasing alleles. A form of selection that is expected to be common for many traits of interest is stabilizing selection, which penalizes deviations from an optimal trait value. By favoring compensating combinations of trait-increasing and trait-decreasing alleles, stabilizing selection generates negative correlations among alleles with the same directional effect [10,11], and therefore can confound effect-size estimates in a GWAS performed on the trait under selection or on a genetically correlated trait.

The potential for dynastic, environmental, and genetic confounds to bias GWAS effect-size estimates has long been recognized [4,12], and so a major focus of the literature has been to develop methods to control for these confounds [13,14]. Standard approaches include using estimates of genetic relatedness as covariates in GWAS regressions [15,16] or downstream analyses such as LD-Score regression [17–19]. Such methods aim to control for both environmental and genetic confounding, but do so imperfectly (e.g., [20,21]). Further, it is often unclear what features of genetic stratification are being addressed [1,2]: assortative mating in particular may not be well accounted for by these methods [22]. Additionally, dynastic confounding will be particularly difficult to control for using population-level covariates.

One promising way forward is to estimate allelic effects within families, either by comparing the separate associations of parentally transmitted and untransmitted alleles with trait values in the offspring [23–27], or by associating differences in siblings' trait values with differences in the alleles they inherited from their parents [28–30]. The idea is that, by controlling for parental genotypes, within-family association studies control for both environmental

stratification and indirect/dynastic effects, while mendelian segregation randomizes alleles across genetic backgrounds. In principle, this allows the "direct genetic effect" of an allele—the causal effect of an allele carried by an individual on their trait value—to be estimated. Recognizing that a variant detected in a GWAS will usually not itself be causal for the trait variation but instead will only be correlated with true causal variants, the direct effect of a genotyped variant is usually interpreted as reflecting the direct causal effects of nearby loci that are genetically correlated with the focal locus [2]—but not the effects of more distant loci that might also be correlated with the focal genotyped locus (e.g., because of population structure or assortative mating).

Consistent with both the presence of substantial confounds in some population-based GWASs and the mitigation of these confounds in within-family GWASs, family-based estimates of direct effect sizes and aggregate quantities based on these estimates (e.g., SNP-based heritabilities) are substantially smaller than population GWAS estimates for a number of traits, most notably social and behavioral traits [30–34]. Likewise, estimates of genetic correlations between traits are sometimes substantially reduced when calculated using direct effect estimates from within-family GWASs (e.g., [33]). While some of these findings could reflect the contribution of indirect genetic effects to population GWASs, it is also likely that, at least for some traits, standard controls for population stratification in population GWASs have been insufficient [20,21,34–37].

Our aim in this paper is to study a general model of confounding in GWASs, to generate clear intuition for its influence on estimates of effect sizes in both population- and family-based designs. A number of the issues that we analyze have previously been raised, particularly in the context of population-based GWASs (e.g., [1,2,38,39]); here, we analyze them in a common framework that allows for comparison of multiple sources of confounding in both population and family-based GWASs. There is a large literature on GWASs in nonhuman organisms (e.g., [40–43]). However, although the results and intuition that we derive here apply equally well to human and nonhuman GWASs, we shall interpret them primarily from the perspective of human GWASs, in which the inability to experimentally randomize environments, together with the small effects that investigators hope to detect, makes confounding a particular concern.

To focus on confounding, we assume no G×E and G×G interactions (the effects of G×E and G×G on estimates produced by population- and family-based designs are studied in ref. [44]). We derive expressions for estimators of direct effects in both population and within-family GWASs, as functions of the true direct and indirect effects at a locus and the genetic confounds induced by other loci. In doing so, we find that family-based estimates of direct effects are in fact susceptible to genetic confounding, contrary to standard interpretation. Reassuringly, in many of the models we consider, the resulting biases are likely to be small in humans. We also address a related case: family-based GWAS designs that consider transmitted and untransmitted parental alleles and in which the indirect (or "dynastic") effect of an allele is estimated from its association with the offspring's phenotype when carried by the parent but not transmitted to the offspring. We show that this estimator of indirect effects can be substantially biased by genetic and environmental confounds, in a similar way to population estimates of direct effects. Next, we consider various sources of genetic confounding—assortative mating, population structure, and stabilizing selection on GWAS traits—and how they influence estimates of direct effects in both population and within-family GWASs.

We then turn to sibling indirect effects, which are known to bias estimates of direct effects in sibling-based GWASs [2,34]. We characterize this bias in a simple model and contrast it to the bias caused by sibling indirect effects in a population GWAS.

## 2 Effect-size estimates in association study designs

Our primary focus will be on how genetic confounding can bias the estimation of direct genetic effects. These genetic confounds are due to associations between a genotyped variant at a GWAS locus and causal variants at other loci. As we will see, 2 kinds of association must be distinguished: cis-linkage disequilibrium (cis-LD) and trans-linkage disequilibrium (trans-LD). Genetic variants $A$ and $B$ are in positive cis-LD if, when an individual inherits $A$ from a given parent, the individual is disproportionately likely to inherit $B$ from that parent (Fig 1A). $A$ and $B$ are in positive trans-LD if, when an individual inherits $A$ from one parent, the individual is disproportionately likely to inherit $B$ from the *other* parent (Fig 1B). These covariances have also been called gametic and non-gametic LD, respectively (e.g., [45]). To quantify the degrees of cis-LD and trans-LD, we denote by $D_{ij}$ and $\tilde{D}_{ij}$ the allelic covariances between focal variants at loci $i$ and $j$ in cis and in trans, and we denote by $r_{ij}$ and $\tilde{r}_{ij}$ the analogous allelic correlation coefficients. For some of our results, it will be important to distinguish the LD present in the sample on which the association study is performed and the LD present among the parents of the sample.

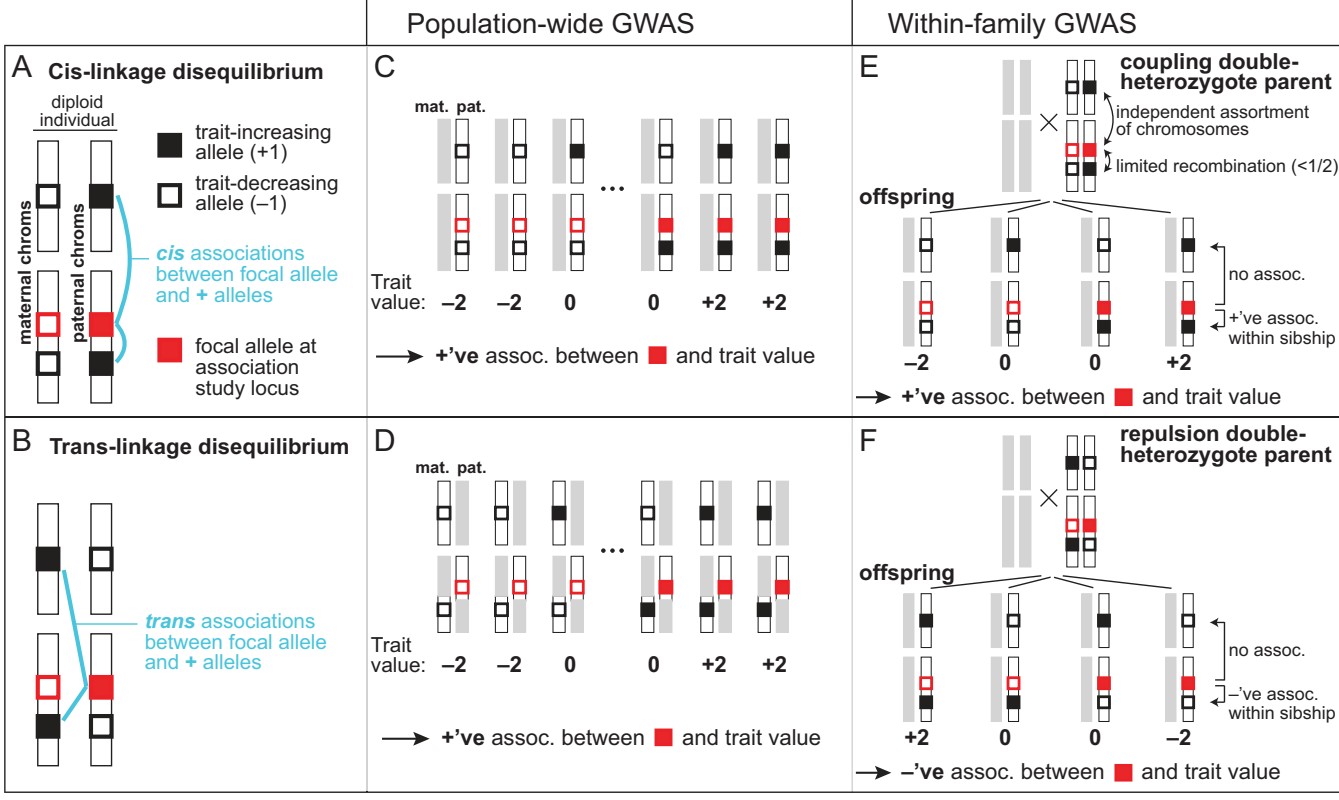

**Fig 1. The influence of cis- and trans-LD on effect-size estimates in population-based and within-family association studies.** (A) The focal allele at an association study locus (solid red square) is in positive cis-LD with trait-increasing alleles at other loci (solid black squares) if it is disproportionately likely to be found alongside them on an individual's maternally or paternally inherited genome. (B) The focal allele at the study locus is in positive trans-LD with trait-increasing alleles at other loci if it is disproportionately likely to be found across from them on the maternally and paternally inherited genomes. (C,D) In a population association study, both positive cis- and trans-LD between the focal allele at the study locus and trait-increasing alleles anywhere else in the genome —either on the same chromosome as the study locus or on different chromosomes—generate a spuriously high effect-size estimate at the study locus. (E,F) In a sibling association study, a trait-increasing allele causes a spuriously increased effect-size estimate at the study locus if the parent is a coupling double heterozygote for the focal and trait-increasing alleles, having inherited them from the same parent (E), but a spuriously decreased estimate if the parent is a repulsion double heterozygote, having inherited them from different parents (F). These biases arise only if the trait-affecting locus is on the same chromosome as the focal study locus. The net bias depends on the relative frequencies of coupling and repulsion double heterozygotes in the parents, which depends on the difference in the degrees of cis- and trans-LD.

Consider a trait Y influenced by genetic variants at a set of polymorphic loci $L$, each of which segregates for 2 alleles. For ease of interpretation, and without loss of generality, we designate the "focal" allele at locus $l \in L$ to be the allele that directly increases the trait value, and we denote by $p_l$ the frequency of this allele. Allelic effects are assumed to be additive within and across loci, such that the trait value of an individual can be written

$$Y = Y^* + \underbrace{\sum_{l \in L} g_l \alpha_l^{\mathrm{d}}}_{\text{direct effects}} + \underbrace{\sum_{l \in L} (g_l^{\mathrm{m}} + g_l^{\mathrm{f}})\alpha_l^{\mathrm{i}}}_{\text{indirect effects}} + \epsilon. \tag{1}$$

Here, $g_l, g_l^{\mathrm{m}}$, and $g_l^{\mathrm{f}}$ are the numbers (0, 1, or 2) of focal alleles carried at locus $l$ by the individual, their mother, and their father, respectively, $\alpha_l^{\mathrm{d}} > 0$ is the direct effect of the focal allele at $l$, and $\alpha_l^{\mathrm{i}}$ is its indirect effect via the maternal and paternal genotypes. (For simplicity, we assume that indirect effects via the maternal and paternal genotypes are equal; this assumption is relaxed in S1 Text Section S1.) $\epsilon$ is the environmental noise, with $\mathbb{E}[\epsilon] = 0$, and $Y^*$ is the expected trait value of the offspring of parents who carry only trait-decreasing alleles.

## 2.1 Population-based association studies

The variants at a genotyped locus will usually not themselves have causal effects on the trait, but will instead be in cis-LD with—and thus "tag"—causal variants at nearby loci. Thus, we typically think of the association at a focal genotyped locus as reflecting the direct contributions of a relatively small number of tightly linked loci, $L_{\mathrm{local}}$, found within tens or perhaps hundreds of kb from the focal locus [46]. For concreteness, we assume some fixed $L_{\mathrm{local}}$ in our analyses. Informally, we intend "local" to capture only those loci that are within a sufficiently short recombination distance of the focal locus that the timescale of recombination between these loci is longer than that by which other processes, such as population structure, assortative mating, and selection, generate LD between more distant loci. We recognize that this definition is somewhat arbitrary, that in practice researchers seldom have a predefined number of "local" SNPs in mind, and that others might prefer alternative definitions (such as the entire chromosome of the focal locus) or to dispense with the idea of "local" altogether. However, we believe our definition is close to what researchers implicitly have in mind when they think of a genotyped locus as "tagging" causal loci. Moreover, many of the processes that we study below generate long-range LD not just within chromosomes but also across different chromosomes; to extend the definition of "tagging" to the entire chromosome of the focal locus, but not to the entire genome, is then also an arbitrary choice.

Under the additive model, the standard interpretation is that a population association study performed at a focal genotyped locus $\lambda$ provides an estimate of the quantity

$$\alpha_\lambda = \frac{1}{p_\lambda(1 - p_\lambda)} \sum_{l \in L_{\mathrm{local}}} D_{\lambda l} \alpha_l^{\mathrm{d}}, \tag{2}$$

where $p_\lambda$ is the frequency of the focal allele at $\lambda$, and $D_{\lambda l}$ is the degree of cis-LD between the focal allele at $\lambda$ and a causal allele at a nearby locus $l \in L_{\mathrm{local}}$. It is reasonable to think of this quantity as the "direct effect" tagged by the focal variant at the genotyped locus $\lambda$: in the absence of confounding, it can be interpreted as the average phenotypic effect of randomly choosing a non-focal allele in the population and swapping it for a focal allele, where in this hypothetical swap, the causal alleles near the locus are included (though see ref. [44] for how G×E and G×G interactions can complicate such causal interpretations).

We shall interpret systematic deviations of effect-size estimates from Eq (2) as "biased." This is because our focus is on the estimation of direct effects at individual loci rather than on

other applications such as trait prediction. For the same reason, and following standard terminology in statistics, we refer to the causes of these biases as "confounds," since they are a form of omitted variable bias in the marginal regression at the study locus $\lambda$. We recognize that these definitions of "bias" and "confounding" might be less appropriate in some other settings.

Effect-size estimation in a population GWAS is complicated by the presence of environmental and genetic stratification. Under the model in Eq (1), if we perform a standard population association study at locus $\lambda$, the estimated effect of the focal allele on the trait Y is

$$\hat{\alpha}_\lambda^{\mathrm{pop}} = \frac{2}{V_\lambda} \left( \sum_{l \in L_{\mathrm{local}}} D_{\lambda l} \alpha_l^{\mathrm{d}} + \underbrace{\sum_{l \in L \setminus L_{\mathrm{local}}} D_{\lambda l} \alpha_l^{\mathrm{d}} + \sum_{l \in L} \tilde{D}_{\lambda l} \alpha_l^{\mathrm{d}}}_{\text{genetic confounds, direct}} + \underbrace{\sum_{l \in L} [D_{\lambda l}' + \tilde{D}_{\lambda l}' + 2\tilde{D}_{\lambda l}] \alpha_l^{\mathrm{i}}}_{\text{genetic confounds, indirect}} + \underbrace{\frac{1}{2} \mathrm{Cov}(g_\lambda, \epsilon)}_{\substack{\text{environmental} \\ \text{confound}}} \right), \quad (3)$$

where, of the cis- and trans-LD terms, $D_{\lambda l}$ and $\tilde{D}_{\lambda l}$ are defined in the GWAS sample while $D_{\lambda l}'$ and $\tilde{D}_{\lambda l}'$ are defined in their parents (S1 Text Section S1.2). $V_\lambda$ is the genotypic variance at $\lambda$, equal to $2p_\lambda(1 - p_\lambda)(1 + F_\lambda)$, where $F_\lambda$ is Wright's coefficient of inbreeding at $\lambda$.

The environmental confound is $\mathrm{Cov}(g_\lambda, \epsilon)/V_\lambda$; all non-local cis- and trans-LD terms in the study sample ($D_{\lambda l}$ and $\tilde{D}_{\lambda l}$, $l \notin L_{\mathrm{local}}$) are direct genetic confounds (Fig 1C and 1D), and all cis- and trans-LD terms among parents of sampled individuals ($D_{\lambda l}'$ and $\tilde{D}_{\lambda l}'$), together with all trans-LD terms in the study sample ($\tilde{D}_{\lambda l}$), are indirect genetic confounds.

The direct genetic confounds arise because an allele carried by an offspring at $\lambda$ is correlated with the alleles that they carry at other loci $l \in L$ (via $D_{\lambda l}$ and $\tilde{D}_{\lambda l}$) that directly affect the trait value. The indirect genetic confounds arise because an allele carried by the offspring at $\lambda$—say, the maternal allele—is correlated with alleles carried by the offspring's mother at other loci ($D_{\lambda l}'$ and $\tilde{D}_{\lambda l}'$) and alleles carried by their father (as reflected by the trans-LD in the offspring, $\tilde{D}_{\lambda l}$). These alleles in the parents can indirectly affect the offspring's trait value.

Thus, as is now well appreciated, population-based GWASs potentially suffer from many types of confounds [1,2]. In practice, they can be reduced by including principal components —which capture genome-wide relatedness among GWAS participants—as regressors in a GWAS, or by using relatedness matrices in mixed models [15,16]. However, it is often unclear exactly what these methods control for in a given application [1,2], they can be sensitive to sample size [47], and they have been shown to be inadequate in important cases (e.g., [20,21]). When principal components (or other controls) fail to account fully for stratification, then Eq (3) can be interpreted as a decomposition of the remaining, uncontrolled-for confounding in the GWAS. (By the Frisch–Waugh–Lovell theorem ([48], pg. 36), Eq (3) is the estimate one obtains by first regressing the focal-locus genotype on the PCs, collecting the residuals from this regressions—which can be thought of as focal-locus genotype values stripped of whatever signal the PCs captured—and regressing the trait value on these residuals.)

## 2.2 Within-family association studies

The 2 within-family association study designs that we consider are parent–offspring GWASs and sibling GWASs. Other designs have been proposed to control for genetic and environmental confounding in the estimation of aggregate quantities such as heritability (e.g., [49]), but our primary focus is on the estimation of single-marker effect sizes. We do later turn to the interpretation of polygenic score (PGS) regressions within families.

**Estimates of direct genetic effects.** Parent–offspring studies can be used to estimate trait associations for parentally transmitted and untransmitted variants at a locus $\lambda$, $\hat{\alpha}_\lambda^{(\mathrm{T})}$, and $\hat{\alpha}_\lambda^{(\mathrm{U})}$,

by regressing the trait value $Y$ jointly on the transmitted and untransmitted genotypes, $g_\lambda^{\mathrm{T}}$ and $g_\lambda^{\mathrm{U}}$ [27] (note that here and throughout, we use "transmitted genotype" and "untransmitted genotype" as shorthand for the genotypes constructed from transmitted and untransmitted alleles, respectively). The aim is often to estimate the direct effect of a variant, $\hat{\alpha}_\lambda^{\mathrm{d}}$, as the difference between these 2 regression coefficients:

$$\hat{\alpha}_\lambda^{\mathrm{d,T-U}} = \hat{\alpha}_\lambda^{(\mathrm{T})} - \hat{\alpha}_\lambda^{(\mathrm{U})}. \tag{4}$$

A second aim is to treat $\hat{\alpha}_\lambda^{(\mathrm{U})}$ as an estimate of the indirect, or family, effect of the variant. We return to this second aim later.

In S1 Text Section S1.4, we show that, in the absence of interactions between parental and offspring genotypes, the estimate of the direct effect of a variant at locus $\lambda$ in a parent–offspring study is

$$\hat{\alpha}_\lambda^{\mathrm{d,T-U}} = \hat{\alpha}_\lambda^{(\mathrm{T})} - \hat{\alpha}_\lambda^{(\mathrm{U})} = \frac{2}{H_\lambda} \sum_{l \in L} (1 - 2c_{\lambda l})(D'_{\lambda l} - \tilde{D}'_{\lambda l})\alpha_l^{\mathrm{d}} \tag{5}$$

$$\approx \frac{2}{H_\lambda} \left( \sum_{l \in L_{\mathrm{local}}} D'_{\lambda l}\alpha_l^{\mathrm{d}} + \underbrace{\sum_{l \in L \backslash L_{\mathrm{local}}} (1 - 2c_{\lambda l})(D'_{\lambda l} - \tilde{D}'_{\lambda l})\alpha_l^{\mathrm{d}}}_{\text{genetic confounds, direct}} \right), \tag{6}$$

where $H_\lambda$ is the fraction of parents who are heterozygous at locus $\lambda$ and $c_{\lambda l}$ is the sex-averaged recombination fraction between $\lambda$ and $l$. The cis- and trans-LD terms $D'_{\lambda l}$ and $\tilde{D}'_{\lambda l}$ are measured in the parents.

Similarly, an estimate of the direct effect can be obtained from pairs of siblings by regressing the differences in their phenotypes on the differences in their genotypes at the focal locus $\lambda$. In the presence of genetic confounds, this procedure yields the same estimate as Eq (6), in expectation (S1 Text Section S1.3):

$$\hat{\alpha}_\lambda^{\mathrm{d,sib}} \approx \frac{2}{H_\lambda} \left( \sum_{l \in L_{\mathrm{local}}} D'_{\lambda l}\alpha_l^{\mathrm{d}} + \underbrace{\sum_{l \in L \backslash L_{\mathrm{local}}} (1 - 2c_{\lambda l})(D'_{\lambda l} - \tilde{D}'_{\lambda l})\alpha_l^{\mathrm{d}}}_{\text{genetic confounds, direct}} \right). \tag{7}$$

An assumption in sibling GWASs is that an offspring's phenotype is not influenced by the genotypes of their siblings—i.e., that there are no sibling indirect genetic effects. We consider violations of this assumption later.

Both of the family-based estimates above implicitly make the further assumption that the association study is conducted before any selection has occurred in the offspring generation, so that the offspring genotypes reflect mendelian transmission from parents. This includes both natural selection and selection in which families/offspring are included in the sample (i.e., ascertainment).

In Eqs (6) and (7), there is no environmental confound, because family-based GWASs successfully randomize the environments of family members with respect to within-family genetic transmission; i.e., variation in offsprings' genotypes around their parental or sibship means is randomly assigned with respect to environment.

The derivations above further show that, while population association studies are biased by sums of trans- and cis-LD between the focal locus and all causal loci (Eq (3)), within-family association studies are instead biased by *differences* between trans- and cis-LD, and moreover, that the biases in within-family studies are driven only by LD between the focal locus and

causal loci on the same chromosome ($c_{\lambda l} < 1/2$). To provide an intuition for this result, we focus our discussion on a sibling association study performed at $\lambda$; the intuition is identical for the analogous parent–offspring study.

Because the difference between 2 siblings in their maternally inherited genotypes is uncorrelated with the difference in their paternally inherited genotypes, we may consider maternal and paternal transmissions separately in studying how a locus $l \in L$ can confound effect-size estimation at $\lambda$ in a sibling association study. We will phrase our discussion in terms of maternal transmission.

For effect-size estimation at $\lambda$ to be genetically confounded by maternal transmission at a distant locus $l$, the mother must be heterozygous at both loci. For if she were homozygous at $l$, then maternal transmission at $l$ could not contribute to any trait differences between her offspring, while if she were homozygous at $\lambda$, maternal transmission would not result in genetic variation among her offspring at $\lambda$ with which trait variation could be associated. Therefore, we restrict our focus to mothers who are heterozygous at both $\lambda$ and $l$, or "double heterozygotes." Two kinds exist (Fig 1E and 1F): coupling double heterozygotes who carry the focal alleles at $\lambda$ and $l$ on the same haploid genome ("in cis") and repulsion double heterozygotes who carry them on opposite haploid genomes ("in trans").

We first consider the case where the recombination rate between $\lambda$ and $l$ is small ($c_{\lambda l} \ll 1/2$). In this case, if the mother is a coupling double heterozygote, then her offspring will tend to inherit either both or neither of the focal alleles at $\lambda$ and $l$ (Fig 1E). Therefore, if one sibling inherits the focal allele at $\lambda$ and another does not, the first sibling will tend to inherit the focal (trait-increasing, as we have defined it) allele at $l$ and the second sibling will not, so that the effect of locus $l$ positively confounds the association between $\lambda$ and the trait (Fig 1E). If the mother is instead a repulsion double heterozygote, then her offspring will tend to inherit either the focal allele at $\lambda$ or the focal allele at $l$, but not both (Fig 1F). In this case, if one sibling inherits the focal allele at $\lambda$ and another does not, the second sibling will tend to inherit the focal (trait-increasing) allele at $l$ and the first sibling will not, so that the effect of locus $l$ negatively confounds the association between $\lambda$ and the trait (Fig 1F). When $\lambda$ and $l$ are linked, therefore, the way in which $l$ genetically confounds the effect-size estimate at $\lambda$ depends, positively or negatively, on whether the fraction of coupling double heterozygotes among parents is greater or smaller, respectively, than the fraction of repulsion double heterozygotes.

In contrast, if $\lambda$ and $l$ are unlinked ($c_{\lambda l} = 1/2$), then transmissions from coupling and repulsion double heterozygote parents are equal, and so $l$ cannot confound estimates at $\lambda$ (Fig 1E and 1F). Put differently, meiosis in double heterozygotes fully randomizes joint allelic transmissions at $\lambda$ and $l$, with offspring equally likely to inherit any possible combination of alleles at the 2 loci.

Therefore, only linked loci $l$ can confound a family-based association study at $\lambda$, and they do so in proportion to (i) how small the recombination rate between $\lambda$ and $l$ is; and (ii) the difference between the fractions of parents who are coupling and repulsion double heterozygotes at $\lambda$ and $l$. Accordingly, if we write these fractions of parents as $H_{\lambda l}^{\text{coup}}$ and $H_{\lambda l}^{\text{rep}}$, then $D'_{\lambda l} - \tilde{D}'_{\lambda l} = (H_{\lambda l}^{\text{coup}} - H_{\lambda l}^{\text{rep}})/2$, and so Eq (7) (and Eq (6)) can be rewritten in terms of the relative frequencies of the 2 kinds of double-heterozygotes:

$$\hat{\alpha}_{\lambda}^{\text{d,sib}} \approx \frac{2}{H_{\lambda}} \left( \sum_{l \in L_{\text{local}}} D'_{\lambda l} \alpha_l^{\text{d}} + \sum_{l \in L \setminus L_{\text{local}}} \left( \frac{1}{2} - c_{\lambda l} \right) (H_{\lambda l}^{\text{coup}} - H_{\lambda l}^{\text{rep}}) \alpha_l^{\text{d}} \right).$$

In a species with many chromosomes, such as humans, for a given locus, there will be many more unlinked loci than linked loci. Therefore, the set of loci that can confound a family-based association study at a given locus will be much smaller than the set of loci that can confound a

population association study at the locus. It will often be the case, therefore, that biases in the estimation of direct genetic effects will be smaller in family-based studies than in population studies, a point that we explore below when we consider sources of genetic confounding.

**Estimates of indirect genetic effects.** We now return to the coefficient on the untransmitted genotype in the joint regression of trait on transmitted and untransmitted genotype in parent–offspring GWASs, $\hat{\alpha}_\lambda^{(U)}$, which has sometimes been treated as an estimate of the indirect effect $\hat{\alpha}_\lambda^i$. Assuming equal indirect effects via maternal and paternal genotypes (an assumption that we relax in S1 Text Section S1.4),

$$\hat{\alpha}_\lambda^i = \hat{\alpha}_\lambda^{(U)}$$

$$= \frac{1}{H_\lambda(1+3F)} \left( 2\sum_{l\in L} \left[ D'_{\lambda l}\Big[(1+3F)c_{\lambda l} - 2F\Big] + \tilde{D}'_{\lambda l}\Big[-(1+3F)c_{\lambda l} + 1 + F\Big] + (1-F)\tilde{D}_{\lambda l}\right]\alpha_l^d \right.$$

$$\left. + 2(1-F)\sum_{l\in L}\left[D'_{\lambda l} + \tilde{D}'_{\lambda l} + 2\tilde{D}_{\lambda l}\right]\alpha_l^i + (1+F)\mathrm{Cov}(g_\lambda^U, \epsilon) - 2F\mathrm{Cov}(g_\lambda, \epsilon) \right). \quad (8)$$

This expression is easiest to interpret when $F = 0$, in which case

$$\hat{\alpha}_\lambda^i = \frac{2}{H_\lambda}\left( \underbrace{\sum_{l\in L_{\text{local}}} D'_{\lambda l}\alpha_l^i}_{\text{local indirect effect}} + \underbrace{\sum_{l\in L\setminus L_{\text{local}}} (D'_{\lambda l}c_{\lambda l} + \tilde{D}'_{\lambda l}(1-c_{\lambda l}) + \tilde{D}_{\lambda l})\alpha_l^d}_{\text{genetic confounds, direct}} + \underbrace{\sum_{l\in L\setminus L_{\text{local}}} (D'_{\lambda l} + \tilde{D}'_{\lambda l} + 2\tilde{D}_{\lambda l})\alpha_l^i}_{\text{genetic confounds, indirect}} + \underbrace{\frac{1}{2}\mathrm{Cov}(g_\lambda^U, \epsilon)}_{\substack{\text{environmental}\\\text{confound}}} \right). \quad (9)$$

(This simplification makes use of the fact that, by our definition of $L_{\text{local}}$, setting $F = 0$ at locus $\lambda$ implies that $\tilde{D}_{\lambda l} = \tilde{D}'_{\lambda l} = 0$ for $l\in L_{\text{local}}$—see Eq S.29 in S1 Text Section S1.4.)

The direct genetic confound reflects associations of the untransmitted alleles at the focal locus with alleles that are transmitted to the offspring at causal loci $l\in L$ and which directly affect the offspring's trait value (via $\alpha_l^d$). These associations are due to covariances among alleles in each parental genome ($D'_{\lambda l}$ and $\tilde{D}'_{\lambda l}$) and across the parental genomes (reflected as trans-LD in the offspring, $\tilde{D}_{\lambda l}$). The indirect genetic confound reflects associations of the untransmitted alleles to alleles at other loci in the parents, which can indirectly affect the offspring trait value (via $\alpha_l^i$). Finally, unlike in family-based estimates of direct genetic effects (Eqs (6) and (7)), family-based estimates of indirect effects suffer from environmental confounding, in the same way that population GWASs do (Eq (3)).

Therefore, using the coefficient on the untransmitted genotype in the joint regression of phenotype on transmitted and untransmitted genotype as an estimate of the indirect effect is highly susceptible to environmental confounding as well as both direct and indirect genetic confounding, in a similar way to estimating the direct effect via a population-based association study [50]. Adjustments for assortative mating in particular have been included in some PGS-based analyses of indirect effects (e.g., [27,34]). However, it is not clear how robust these adjustments are in the presence of multiple forms of confounding.

## 2.3 Polygenic scores and their phenotypic associations

A current drawback to family-based GWASs is that sample sizes are often small, limiting power to estimate direct genetic effects. Because of this limitation, instead of estimating per-locus effect sizes in family designs, investigators often measure the within-family phenotypic

association of a combined linear predictor, a PGS, constructed using effect-size estimates across many loci from a population GWAS. In the sibling-based version of this study design, the difference in siblings' population-based PGSs is regressed on their difference in phenotypes (e.g., [30,31]). In parent–offspring designs, offsprings' trait values are regressed on their population-based PGSs, controlling for the midparent PGSs (e.g., [35]), or, equivalently, offsprings' trait values are regressed jointly on their population-based PGSs constructed separately for transmitted and untransmitted alleles, and the difference in the slopes in this joint regression is taken (e.g., [27]).

When such PGS regressions are used within families for the same phenotype as the population GWAS, a non-zero slope of the PGS is usually interpreted as reflecting the fact that the PGS—despite having been calculated from a population GWAS and therefore subject to many potential confounds—nevertheless does capture the direct genetic effects of alleles. When the PGS for one phenotype is regressed within families on the value of another phenotype, non-zero slopes are often interpreted as evidence that direct genetic effects on the 2 phenotypes are causally related, for example, through pleiotropic effects of the alleles involved.

Suppose that we have performed a population GWAS for trait 1, generating effect-size estimates $\hat{\alpha}_\lambda$ at a set of genotyped loci $\lambda \in \Lambda$. To construct a PGS for trait 1, these effect-size estimates are used as weights in a linear sum across an individual's genotype:

$$PGS_1 = \sum_{\lambda \in \Lambda} g_\lambda \hat{\alpha}_\lambda^{\text{pop}}. \tag{10}$$

In a sibling-based study (the results and intuition below will be the same for a parent–offspring study), the difference between siblings' trait-1 PGSs, $\Delta PGS_1$, is regressed on the difference in their values for trait 2, $\Delta Y_2$ (note that trait 2 could be the same as trait 1). If $L$ is the set of loci that causally underlie variation in trait 2, and $\beta_l$ are the true direct genetic effects of variants at these loci on trait 2, then the numerator of the slope in this regression can be written as follows:

$$\text{Cov}(\Delta PGS_1, \Delta Y_2) = 2 \sum_{\lambda \in \Lambda} \sum_{l \in L} (1 - 2c_{\lambda l})(D'_{\lambda l} - \tilde{D}'_{\lambda l}) \hat{\alpha}_\lambda^{\text{pop}} \beta_l \tag{11}$$

(see S1 Text Section S2). Note that, while the population-based effect-size estimates $\hat{\alpha}_\lambda$ depend on cis- and trans-LD, as detailed by Eq (3), the patterns of LD may differ from those in the family study (the $D'_{\lambda l} - \tilde{D}'_{\lambda l}$ term in Eq (11)) if the population- and family-based studies differ in relevant aspects of sample composition.

The intuition for Eq (11) is similar to that for the single-locus effect-size estimate in a sibling GWAS (Eq (7)). The numerator of the difference in slopes of transmitted and untransmitted PGSs in a parent–offspring design takes a similar form to Eq (11).

In the absence of confounding and under some simplifying assumptions, the sibling PGS covariance measures the contribution of each locus included in the PGS to the additive genic covariance between traits 1 and 2 that is tagged by the genotyped variants included in the PGS (see Eq S.38 in S1 Text Section S2). Under these assumptions, the sibling PGS slope therefore does provide a measure of the underlying pleiotropy between the traits.

Interpretation of the sibling PGS slopes is more complicated in the presence of genetic confounding (see Eq S.37 in S1 Text Section S2), which is absorbed into the effect-size estimates $\hat{\alpha}_\lambda^{\text{pop}}$ (Eq (3)) so that the PGS applies a potentially strange set of weights to the genotyped loci it includes. (A related problem occurs when indirect genetic effects absorbed by the population-based PGS change the interpretation of within-family PGS slopes [51,52].) A non-zero sibling PGS slope still establishes that the trait-1 PGS loci are in systematic signed intra-chromosomal

LD with loci that causally affect trait 2. However, it no longer necessarily implies that traits 1 and 2 are causally related via pleiotropy, for 2 reasons. To understand these reasons, suppose that the causal loci for traits 1 and 2 are distinct, i.e., that there is in fact no pleiotropy. First, a SNP included in the trait-1 PGS could tag local variants that causally affect trait 1 but which are also, via sources of confounding such as cross-trait assortative mating, in systematic long-range LD with variants on the same chromosome that causally affect trait 2. Such SNPs will be predictive of sibling differences in trait 2, even though they locally tag only trait-1 causal variants. Second, LD between variants on the same or distinct chromosomes that are causal for trait 1 and trait 2 will cause some SNPs that locally tag trait-2 causal variants to be significantly associated with trait 1 in a population GWAS, and therefore to be included in the trait-1 PGS. These SNPs, since they tag trait-2 causal variants, will be predictive of sibling differences in trait 2.

In summary, in the presence of confounding, non-zero sibling PGS slopes cannot be viewed as de facto evidence for causal relationships between traits.

## 3 Sources of genetic confounding in association studies

As we have seen, genetic confounding of association studies depends, in ways that vary across study designs, on levels of non-local cis- and trans-LD between the study locus and loci that influence the study trait. Below, we consider various processes that give rise to non-local cis- and trans-LD, and their likely impact on the different association study designs. We focus our attention on the potential for these sources of LD to confound measurement of several key metrics. First, the average deviation of the estimated effect size from its true value, $\mathbb{E}[\hat{\alpha}_\lambda - \alpha_\lambda]$. This metric indicates if effect sizes are systematically overestimated or underestimated because of genetic confounding. Second, the average squared effect-size estimate, weighted by heterozygosity: $\mathbb{E}[2p_\lambda(1-p_\lambda)\hat{\alpha}_\lambda^2]$. The quantity $2p_\lambda(1-p_\lambda)\hat{\alpha}_\lambda^2$ is proportional to the $\chi^2$-statistic of the association test at $\lambda$ and determines whether the detected association is statistically significant. Its expectation is related to important measures such as the genetic variance and SNP-based heritability, and to LD-score regression [18] (we explore the connection of our results to LD-score regression in S1 Text Section S4). It is also directly related to the variance of effect-size estimates, and therefore captures the additional noise that genetic confounding creates in effect-size estimation at a given locus. Third, if GWASs have been performed on more than one trait, the covariance across loci of the effect-size estimates for 2 traits may be of interest. This covariance is determined by the average heterozygosity-weighted product $\mathbb{E}[2p_\lambda(1-p_\lambda)\hat{\alpha}_\lambda\hat{\beta}_\lambda]$, where $\hat{\alpha}_\lambda$ and $\hat{\beta}_\lambda$ are the effect-size estimates at locus $\lambda$ for traits 1 and 2.

In what follows, for simplicity, we ignore indirect effects and assume that there is no environmental confounding (i.e., no correlation between genotypes and the environmental effects $\epsilon$). For each of the sources of genetic confounding that we consider, we calculate the 3 measures listed above both analytically and in whole-genome simulations carried out in SLiM 4.0 [53]. In our simulations, we use 2 recombination maps: (i) for illustrative purposes, a simple hypothetical map where the genome lies along a single chromosome of length 1 Morgan; and (ii) the human linkage map generated by Kong and colleagues [54]. A more detailed description of the simulations can be found in the Methods, and code is available at https://doi.org/10.5281/zenodo.10520811 and https://github.com/cveller/confoundedGWAS.

### 3.1 Assortative mating

Assortative mating is the tendency for mating pairs to be correlated for particular traits—either the same trait (same-trait assortative mating) or distinct traits (cross-trait assortative mating). For example, humans are known to exhibit same-trait assortative mating for height and cross-

trait assortative mating for educational attainment and height (among many other examples, reviewed in refs. [37,55]). Assortative mating generates both cis- and trans-LD: It generates positive trans-LD among trait-increasing alleles because genetic correlations between mates translate to genetic correlations between maternally and paternally inherited genomes, and it generates positive cis-LD among trait-increasing alleles because, over generations, recombination converts trans-LD into cis-LD [9]. (In some cases, assortative mating can generate cis-LD by mechanisms additional to recombination—see ref. [56].)

**Constant-strength assortative mating.** If the strength of assortative mating, measured by the phenotypic correlation among mates $\rho$, is constant over time and there are no other sources of genetic confounding such as population structure, then, for a given pair of loci $l$, $l' \in L$, the positive cis-LD $D_{ll'}$ will initially be smaller than the positive trans-LD $\tilde{D}_{ll'}$, but will gradually grow towards an equilibrium value equal to the trans-LD ($D_{ll'}^* = \tilde{D}_{ll'}^*$); in this equilibrium, assortative mating generates new cis-LD at the same rate as old cis-LD is destroyed by recombination (S1 Text Section S3.1; [9]).

Therefore, in a population GWAS, effect-size estimates will initially be biased upwards because of positive trans-LD, and the magnitude of the bias will grow over time as positive cis-LD too is generated from this trans-LD (Eq (3); Fig 2). In contrast, in a family-based GWAS, effect-size estimates will initially be biased downwards because the positive trans-LD exceeds the positive cis-LD (Eqs (6) and (7); Fig 2). However, as the cis-LD grows over time towards the value of the trans-LD, the magnitude of the downward bias will shrink, and, in equilibrium, the family-based GWAS will not be confounded by assortative mating (Fig 2).

Under certain simplifying assumptions, we can calculate the average bias that assortative mating induces in a population GWAS in equilibrium, in the absence of other sources of genetic confounding such as population structure (S1 Text Section S3.1). In the case of same-trait assortative mating, effect-size estimates are inflated by an average factor of approximately

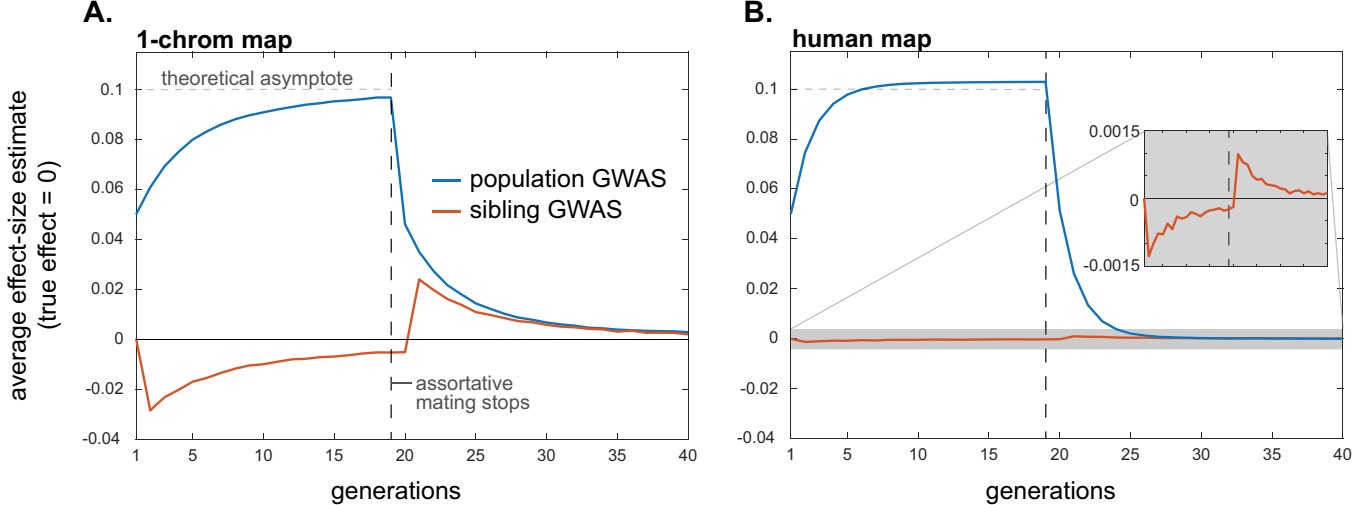

**Fig 2. Assortative mating systematically biases effect-size estimation in population and within-family GWASs, although the bias in within-family GWASs is expected usually to be small.** Here, cross-trait assortative mating between traits 1 and 2 occurs among parents for the first 19 generations, after which mating is random. Assortative mating is sex-asymmetric, with strength $\rho$ = 0.2. Distinct sets of loci underlie variation in trait 1 and 2, with effect sizes at causal loci normalized to 1. Plotted are average estimated effects of the focal alleles at loci causal for trait 1 in population and within-family GWASs on trait 2, for a hypothetical genome with 1 chromosome of length 1 Morgan (A) and for humans (B). Since the traits have distinct genetic bases, the true effects on trait 2 of the alleles at trait-1 loci are zero. The horizontal lines at 0.1 are a theoretical "first-order" approximation of the asymptotic bias in a population GWAS (S1 Text Section S3.1). Profiles are averages across 10,000 replicate simulation trials. Simulation details can be found in the Methods and the code can be found at https://doi.org/10.5281/zenodo.10520811.

$1/(1-h^2\rho)$, where $\rho$ is the phenotypic correlation among mates and $h^2$ is the trait heritability (for similar calculations, see refs. [22,57]). As an example, the strength of assortative mating for human height has been estimated as $\rho\sim0.25$ [58], which, together with a heritability of $h^2\sim0.8$, implies that effect-size estimates in a population-based GWAS would be inflated by a factor of about $1/(1-h^2\rho)\sim1.25$, a 25% amplification.

In the case of cross-trait assortative mating, if assortative mating is directional/asymmetric with respect to sex—i.e., the correlation $\rho$ is between female trait 1 and male trait 2—then assortative mating generates spurious associations between trait 1 and alleles that affect trait 2 (and vice versa). If the loci underlying the 2 traits are distinct, then, in equilibrium, the spurious effect-size estimate at non-causal loci is approximately $h^2\rho/2$ times the effect at causal loci, assuming the traits to have the same heritabilities and genetic architectures (horizontal dashed line in Fig 2; see S1 Text Section S3.1 for relaxations of these assumptions). If cross-trait assortative mating is bidirectional/symmetric with respect to sex, then, in equilibrium, the average spurious effect-size estimate at non-causal loci is approximately $h^2\rho$ times the effect at causal loci. Upward biases in effect-size estimates at causal loci are also expected under cross-trait assortative mating, but these are second-order relative to the biases at non-causal loci (S1 Fig).

The systematic over- and underestimation of effect sizes that assortative mating induces in population and family-based GWASs, respectively, will also affect our second measure of interest, the heterozygosity-weighted average squared effect-size estimate $\mathbb{E}[2p_\lambda(1-p_\lambda)\hat{\alpha}_\lambda^2]$ (and therefore, also downstream quantities such as SNP heritabilities). In a population GWAS, the presence of trans-LD and the gradual creation of cis-LD under assortative mating will increase the biases in effect-size estimates over time (Fig 2), which will concomitantly increase the average value of $\hat{\alpha}_\lambda^2$ (Fig 3; also see ref. [22]). Moreover, cross-trait assortative mating will generate signals of genetic correlations among traits even in the absence of any pleiotropic effects of underlying variants [37]. In a family-based GWAS, the temporary attenuation of effect-size

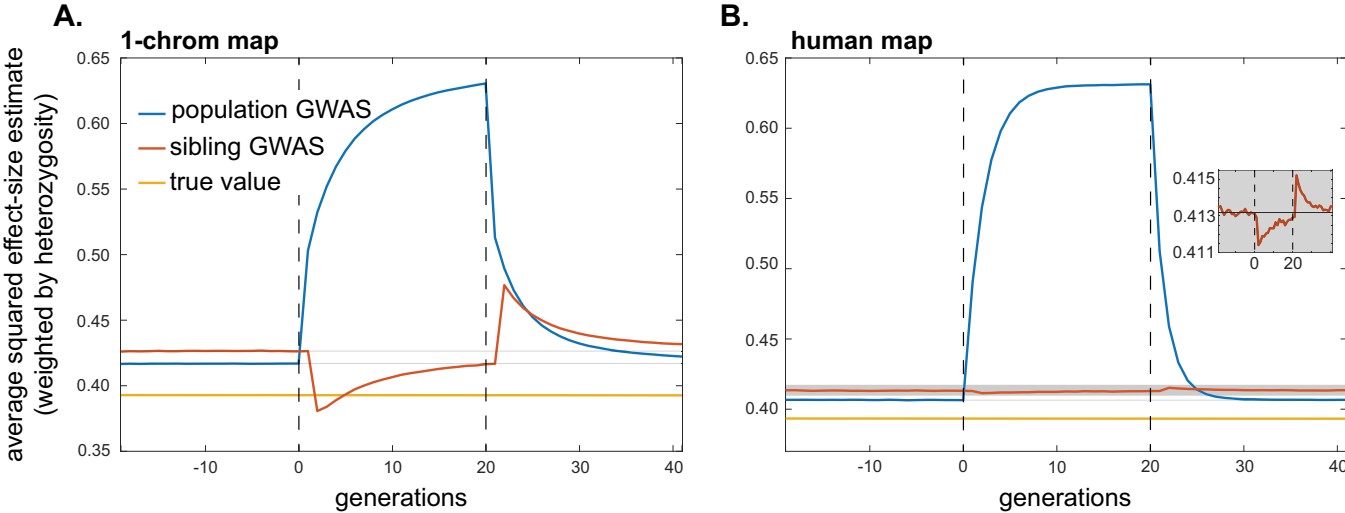

**Fig 3. The impact of assortative mating on the average squared effect-size estimate in population and within-family GWASs.** Same-trait assortative mating of strength $\rho = 0.2$ occurs among parents in generations 0–20; mating is random before and after this period. Under random mating, the average squared effect-size estimates exceed the true average squared effect size (yellow line) because random drift generates chance local LD with causal alleles that inflates the variance of effect-size estimation (e.g., [18]). The magnitude of this variance inflation depends on the GWAS design and sample size, and the effect of assortative mating and its cessation should be judged in reference to it. To guide the eye in this judgment, the faint horizontal lines in (A) and (B) show the average squared effect-size estimate in the last 20 generations of the initial burn-in period of random mating. The inset in (B) zooms in on the sibling GWAS profile, omitting the population GWAS profile for clarity. Profiles are averages across 5,000 replicate simulation trials. Simulation details can be found in the Methods and the code can be found at https://doi.org/10.5281/zenodo.10520811.

estimates owing to a transient excess of trans-LD over cis-LD under assortative mating will lead to a similar attenuation in the average squared effect-size estimate (Fig 3), although, like the bias in effect-size estimates themselves, this attenuation is expected to be small in humans (Fig 3B).

As shown by Border and colleagues [22,37], the effects of assortative mating on estimates of heritability and genetic correlations described above are not well controlled for by LD Score regression [17,18]. The LD score of a variant proxies the amount of local causal variation the SNP tags, but because assortative mating generates long-range signed LD among causal variants, it causes local causal variants to be in long-range signed LD with other causal variants throughout the genome. Therefore, the slope of the LD score regression absorbs the effects of assortative mating, causing its estimates of heritability and of the degree of pleiotropy to be inflated (S1 Text Section S4). We note that the definition of heritability itself, and its measurement using various study designs, can be complicated by processes such as assortative mating (e.g., [59]).

**Historical assortative mating.** If, at some point in time, assortative mating for traits ceases and mating becomes random with respect to those traits, the positive trans-LD that was present under assortative mating will immediately disappear, leaving only the positive cis-LD that had built up; this cis-LD will then be gradually eroded by recombination. If equilibrium had been attained under assortative mating, the cis-LD would have grown to match the per-generation trans-LD. Therefore, in the first generation after assortative mating ceases, the upward bias in population-GWAS effect-size estimates would halve as the trans-LD disappears (Eq (3)); the bias would then shrink gradually to zero as the cis-LD erodes (Fig 2). A similar pattern will be observed for the heterozygosity-weighted average value of $\hat{\alpha}_\lambda^2$ in the population GWAS, which eventually returns to its equilibrium level under random mating (Fig 3).

In contrast, with the disappearance of the positive trans-LD but the persistence of positive cis-LD, the bias in family-based effect-size estimates will suddenly become positive once assortative mating ceases (having temporarily been negative under assortative mating before equilibrium was attained); this bias too will then gradually shrink to zero as recombination erodes the remaining cis-LD (Fig 2). Concomitantly, the average squared effect-size estimate in the family GWAS will suddenly increase when assortative mating ceases, after which it too will gradually return to its equilibrium value under random mating (Fig 3).

**Assortative mating between traits with different genetic architectures.** An important practical question is how genetic confounding affects the GWAS loci we prioritize for functional follow-up and for use in the construction of PGSs. SNPs are usually prioritized on the basis of their GWAS $p$-value, which is proportional to the estimated variance explained by a SNP, $2p_\lambda(1 - p_\lambda)\hat{\alpha}_\lambda^2$ (where $p_\lambda$ is the minor allele frequency). The results above assume, in the case of cross-trait assortative mating, that the traits involved have similar genetic architectures (distribution of $p_l$ and $\alpha_l$ at causal loci, and the total number of causal loci). In that case, if there is no pleiotropy between the traits, then while SNPs that tag trait-1 causal loci are predictive of the value of trait 2 owing to LD between trait-1 and trait-2 causal loci, we nonetheless expect the SNPs that tag trait-2 causal loci to be better predictors of trait 2, such that GWAS investigators would primarily pick out SNPs tagging trait-2 causal loci for prioritization and use in PGSs.

However, analysis of human GWASs suggests that quantitative traits can have widely different genetic architectures, with, in particular, substantial differences in the effective numbers of causal loci involved and in the distribution of minor allele frequencies ([60] and references therein). If 2 traits with distinct genetic bases show cross-trait assortative mating, but trait 1 has a denser genetic architecture (fewer causal loci) than trait 2, then the genetic signal of

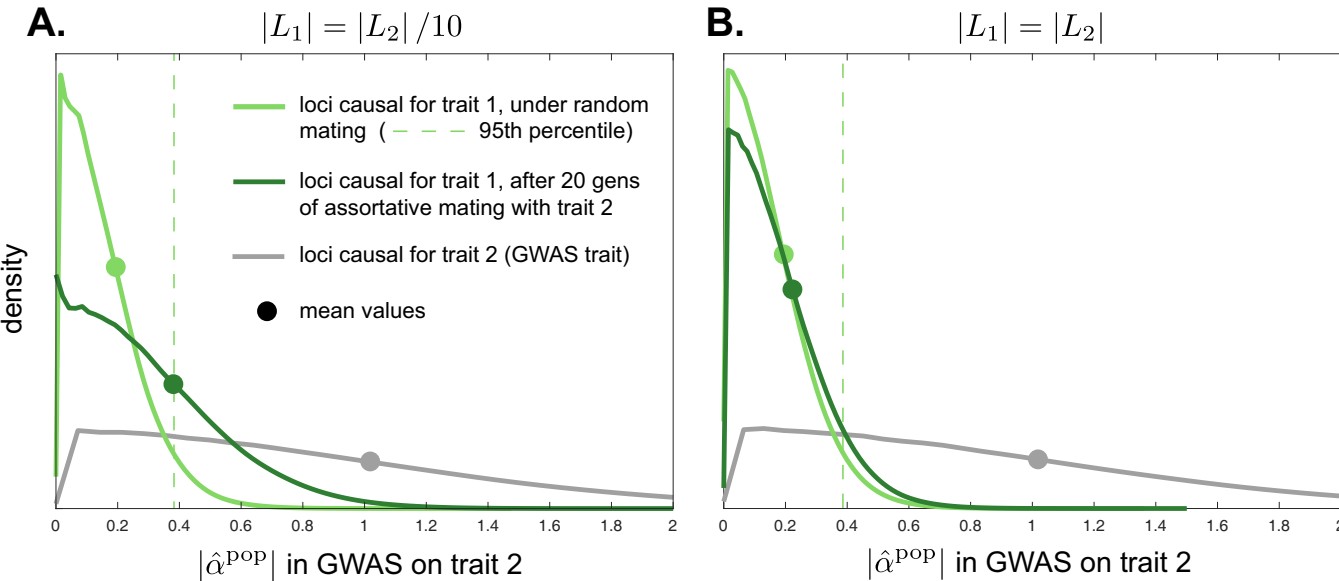

**Fig 4. Cross-trait assortative mating for traits with different genetic architectures can generate large spurious effect-size estimates in population GWASs.**
Shown, for a population GWAS on trait 2, are estimated distributions of the magnitude of effect-size estimates at loci causal for trait 2 (gray) and at loci causal for trait 1 (greens), under random mating (light green) and after 20 generations of cross-trait assortative mating (sex-asymmetric, of strength $\rho = 0.2$) for traits 1 and 2 (dark green). Although the true effects of trait-1 loci on trait 2 are zero in these simulations (no pleiotropy), there is sampling noise in effect-size estimation at trait-1 loci under random mating (light green line), so that the mean magnitude of effect-size estimates is shifted away from zero (light green dot; dashed line displays 95th percentile under random mating). In (A), the traits have equal heritability, but the number of loci contributing variation to trait 1 is 10-fold smaller than that for trait 2. Under assortative mating, the magnitudes of the spurious effect-size estimates at trait-1 loci shift significantly rightward (dark green line), coming to overlap substantially with the distribution of effect-size estimate magnitudes at causal trait-2 loci (gray line; the distribution for trait-2 loci does not appreciably differ under random and assortative mating). In (B), equal numbers of loci contribute variation to the 2 traits. In this case, the rightward shift of the distribution of effect-size estimate magnitudes at trait-1 loci is not as pronounced as in (A). Densities are estimated from pooled effect-size estimates from 1,000 replicate simulations. Simulation details in Methods and the code can be found at https://doi.org/10.5281/zenodo.10520811.

assortative mating—systematic LD between trait-1 and trait-2 causal loci—will be more heavily loaded per-locus onto trait-1 loci than onto trait-2 loci. In a GWAS on trait 2, this will inflate the magnitude of spurious effect-size estimates at SNPs that tag trait-1 loci relative to effect-size estimates at SNPs that tag causal trait-2 loci. In S1 Text Section S3.1, we quantify this effect, showing that, in a population GWAS for trait 2, the average magnitude of spurious effect-size estimates at trait-1 loci is proportional to $\sqrt{|L_2|/|L_1|}$, where $|L_1|$ and $|L_2|$ are the numbers of loci underlying variation in traits 1 and 2, respectively. Thus, when trait 1 has a denser genetic architecture than trait 2 ($|L_2|/|L_1|$ is large), the magnitudes of effect-size estimates at non-causal trait-1 loci could substantially overlap with those at causal trait-2 loci (as illustrated in Fig 4), potentially causing part of the apparent, mappable genetic architecture of the trait-2 GWAS to actually tag trait-1 loci.

### 3.2 Population structure

When a population GWAS draws samples from individuals of dissimilar ancestries, differences in the distribution of causal genotypes, and potentially of environmental exposures, can confound the association study [1,4]. Correcting for confounds due to population structure has therefore been an important pursuit in the GWAS literature [14,23,61].

For concreteness, consider a simple model where 2 populations diverged recently, with no subsequent gene flow between them. Genetic drift—and possibly selection—in the 2 populations will have led to allele frequency differences between them at individual loci. If allele frequencies have diverged at both a genotyped study locus and at loci that causally influence the

study trait, these frequency differences will manifest as linkage disequilibria between the study locus and the causal loci in a sample taken across both populations, even if the loci are not in LD within either population. Specifically, if the frequencies of the focal allele at a given locus $k$ are $p_k^{(1)}$ and $p_k^{(2)}$ in populations 1 and 2, then the cis-LD between the focal alleles at the association study locus $\lambda$ and a causal locus $l$ is

$$D_{\lambda l}^{(S)} = \frac{1}{4}\left(p_\lambda^{(1)} - p_\lambda^{(2)}\right)\left(p_l^{(1)} - p_l^{(2)}\right) \tag{12}$$

in a sample that weights the 2 populations equally, with the superscript *(S)* denoting that this LD is due to stratification. The trans-LD takes exactly the same form: $\tilde{D}_{\lambda l}^{(S)} = D_{\lambda l}^{(S)}$. From Eq (3), locus $l$ therefore confounds estimation of the direct effect at $\lambda$ in a population GWAS, by an amount proportional to

$$2(D_{\lambda l}^{(S)} + \tilde{D}_{\lambda l}^{(S)})\alpha_l^{\mathrm{d}} = (p_\lambda^{(1)} - p_\lambda^{(2)})(p_l^{(1)} - p_l^{(2)})\alpha_l^{\mathrm{d}}. \tag{13}$$

These genetic confounds are in addition to environmental confounding that would arise if the environments of the 2 populations alter their average trait values by different amounts.

In contrast, estimates of direct effects obtained from within-family association studies are not genetically confounded, because cis- and trans-LD are equal (Eqs (6) and (7)). Another way of seeing this is to consider that, by controlling for family, within-family GWASs control for the population, and in the scenario considered, by construction, there are no within-population LDs to confound effect-size estimation.

**Allele frequency divergence due to drift.**   How do the confounds introduced by population structure affect the first of our measures of interest, the average deviation of effect-size estimates from their true values? The answer depends on the source of allele frequency differences between the 2 populations. If the differences are due to neutral genetic drift, they will be independent of each other (assuming causal loci are sufficiently widely spaced) and independent of the direction and size of effects at individual loci. Therefore, the LD induced by these allele frequency differences will, on average, not bias effect-size estimates in a population GWAS:

$$\mathbb{E}[(p_\lambda^{(1)} - p_\lambda^{(2)})(p_l^{(1)} - p_l^{(2)})\alpha_l^{\mathrm{d}}] = \mathbb{E}[p_\lambda^{(1)} - p_\lambda^{(2)}]\mathbb{E}[p_l^{(1)} - p_l^{(2)}]\mathbb{E}[\alpha_l^{\mathrm{d}}] = 0, \tag{14}$$

since $\mathbb{E}[p_k^{(1)} - p_k^{(2)}] = 0$ at any locus $k$.

However, the LD induced by population structure will inflate the average squared effect-size estimate, and by extension the variance of effect-size estimates (Fig 5). In S1 Text Section S3.2, we quantify this effect for the same simple case of 2 separate populations. We find that the average squared effect-size estimate in a population GWAS is an increasing function of the divergence between the 2 populations (as measured by $F_{ST}$), the number of loci contributing variation to the study trait, and the true average squared effect size per locus (see also refs. [38,62]).

In contrast, because effect-size estimates from within-family GWASs are not confounded in this model of isolated populations, the average squared effect-size estimate will not differ substantially from its expectation in an unstructured population (Fig 5).

While we have focused on a simple model of 2 isolated populations, the result that within-family association studies are not confounded holds for other kinds of population structure as well. Specifically, we may be concerned that a population GWAS suffers from genetic confounding along some given axis of population stratification. However, the family-based estimates will be unbiased by confounding along such an axis if the maternal and paternal genotypes at each locus are exchangeable with respect to each other along this axis (S1 Text

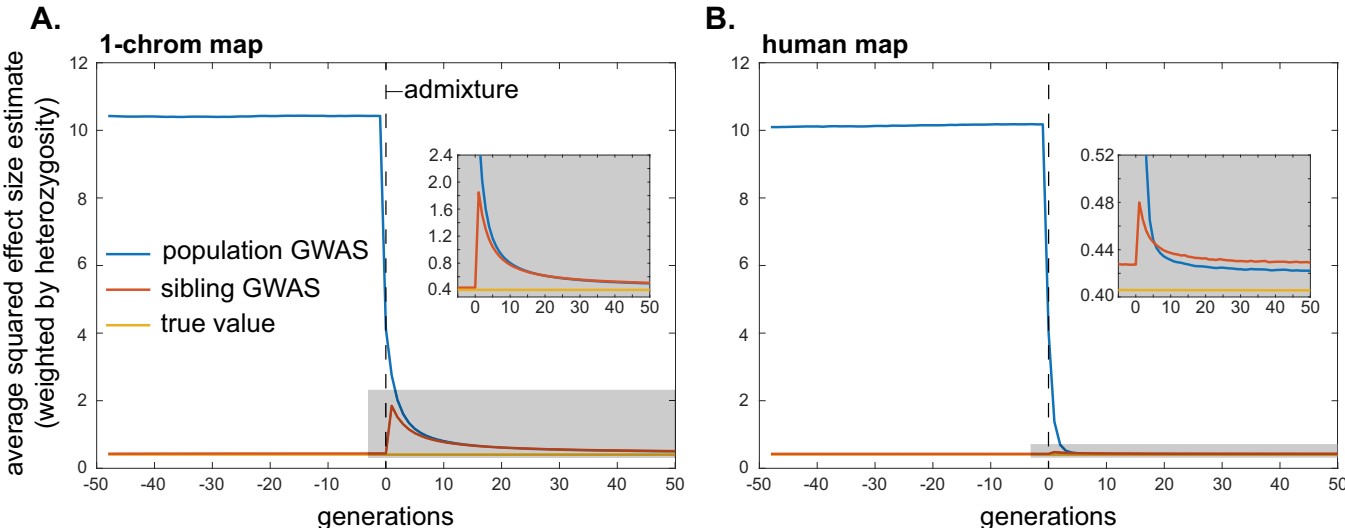

**Fig 5. The impact of population structure and admixture on the average squared effect-size estimate in population and within-family GWASs.** Here, 2 populations are isolated until generation 0, at which point they mix in equal proportions. Initial allele frequencies are chosen independently for the 2 populations, such that allele frequency differences between the populations resemble those that would accumulate over time via random drift. As in Fig 3, the equilibrium value of the mean squared effect-size estimate under random mating is greater than the true mean squared effect size, in both the population and within-family GWAS, owing to linkage disequilibria among causal alleles that arise due to drift. This explains why, in the insets, the blue (population) and red (within-family) profiles do not shrink all the way down to the yellow (true) line after admixture, when mating is random. Note too the difference in scale of the y-axes in the insets: the return to equilibrium is much more rapid under the human genetic map (B) than for a hypothetical genome of one chromosome of length 1 Morgan (A), since, with more recombination, the ancestry-based linkage disequilibria are broken down more rapidly. Profiles are averages across 10,000 replicate simulation trials. Simulation details can be found in the Methods and the code can be found at https://doi.org/10.5281/zenodo.10520811.

Section S3.2). This requirement will be met in expectation under many models of local genetic drift in discrete populations or along geographic gradients. However, as we will shortly argue, migration and admixture introduce further complications.

**Allele frequency divergence due to selection or phenotype-biased migration.** Selection and phenotype-biased migration can also generate allele frequency differences among populations (for a review of phenotype-biased migration, see ref. [63]). Unlike genetic drift, both of these forces can lead to systematic directional associations between effect sizes and changes in allele frequencies between populations. For example, if selection has favored alleles that increase the trait in population 1 but not in population 2, then

$$\mathbb{E}[(p_l^{(1)} - p_l^{(2)})\alpha_l^{d}] > 0 \tag{15}$$

as directional selection causes systematic changes in allele frequencies across the loci $l$ underlying variation in the trait under selection (e.g., [64]). Importantly, this form of selection can occur even if the mean phenotype of the 2 populations does not change [65,66]. Similarly, phenotype-biased migration, where, say, individuals with a higher value of the phenotype tend to migrate from population 2 to population 1, can also create a positive association between effect sizes and allele frequency differences (Eq (15)).

Unlike the case of neutral genetic drift in the 2 populations, where the sign of the LD between 2 alleles is independent of their effect sizes, the effect-size-correlated associations driven by selection or phenotype-biased migration can add up across loci, and thus lead to substantial, systematic biases in estimates of allelic effect sizes. This systematic genetic confounding would also substantially inflate the average squared effect-size estimate and thus measures of the genetic variance tagged by SNPs.

In addition, these systematic sources of genetic confounding can generate genetic correlations between traits with no overlap in their sets of causal loci—i.e., with no pleiotropic relationship. This will occur if 2 traits have both experienced selection or biased migration along the same axis. To take a concrete example, if people tend to migrate to cities in part based on traits 1 and 2, then these traits will become genetically correlated. If this axis is explicitly included as a covariate in the GWAS, then its influence on estimates of heritability and genetic correlations will be removed. However, its influence will not be removed by inclusion of genetic principal components or the relatedness matrix, if this axis (here, city versus non-city) is not a major determinant of genome-wide relatedness at non-causal loci [1]. Nor will LD score regression control for this influence, as the selection- or migration-driven differentiation of a variant along the axis will be correlated with the extent to which it tags long-range causal variants involved in either trait. This effect on LD score regression is similar to that discussed above for assortative mating [22,37]. Thus, like assortative mating, selection and phenotype-biased migration along unaccounted-for axes of population stratification can generate genetic correlations between traits. These selection- and migration-driven correlations should not necessarily be viewed as spurious, since genetic correlations should include those that arise from systematic long-range LD, but they complicate the interpretation of population-level genetic correlations as evidence for pleiotropy.

Again, these issues largely vanish in family-based studies, although phenotype-biased migration can cause transient differences in cis- and trans-LD that lead to biases in family-based estimates of direct effects (Eqs (6) and (7)).

## 3.3 Admixture

When populations that have previously been separated come into contact, alleles from the same ancestral population remain associated with each other in the admixed population until they are dissociated by recombination. If allele frequencies had diverged between the ancestral populations, this "ancestry disequilibrium" can translate to cis-LD between loci affecting a trait [67], potentially confounding GWASs performed in the admixed population. More generally, long-range LD will be an issue when there is genetic stratification and ongoing migration between somewhat genetically distinct groups.

For concreteness, we again consider a simple model where 2 populations have been separated for some time, allowing allele frequencies to diverge between them. The populations then come into contact and admix in the proportions $A$ and $1-A$. We assume that mating is random with respect to ancestry in the admixed population.

Suppose that, just before admixture, the frequencies of the focal allele at a given locus $k$ were $p_k^{(1)}$ and $p_k^{(2)}$ in the 2 populations. Then, the initial degree of cis-LD between loci $\lambda$ and $l$ in the admixed population is given by Eq (12), weighted by the proportions in which the populations admix:

$$D_{\lambda l,0}^{(A)} = A(1-A)(p_\lambda^{(1)} - p_\lambda^{(2)})(p_l^{(1)} - p_l^{(2)}); \tag{16}$$

see, e.g., ref. [68]. This cis-LD subsequently decays at a rate $c_{\lambda l}$ per generation, so that, $t$ generations after admixture,

$$D_{\lambda l,t}^{(A)} = D_{\lambda l,0}^{(A)}(1 - c_{\lambda l})^t = A(1-A)(p_\lambda^{(1)} - p_\lambda^{(2)})(p_l^{(1)} - p_l^{(2)})(1 - c_{\lambda l})^t. \tag{17}$$

Because we assume that mating is random in the admixed population, the trans-LD is zero in every generation after admixture: $\tilde{D}_{\lambda l,t}^{(A)} = 0$. Note that the decay of cis-LD in an admixed population will be slowed if individuals mate assortatively by ancestry, because the trans-LD

generated by assortative mating is continually converted by recombination to new cis-LD (as in our assortative mating model above; see ref. [69] for more discussion of this point in the context of population admixture).

**Allele frequency divergence due to drift.**   How do these patterns of LD affect a population GWAS? If allele frequency differences between populations arose from neutral drift, they will be independent of effect sizes at causal loci and across loci, and therefore will not contribute, on average, a systematic directional bias to effect-size estimates. However, they will inflate the average squared effect-size estimate, by a smaller amount than for a population GWAS performed when the populations were still separated (because of the elimination of trans-LD under random mating in the admixed population). Moreover, this amount will decline in the generations after admixture as the remaining cis-LD is eroded by recombination (Eq (17); Fig 5). We quantify these effects in S1 Text Section S3.3 (see also refs. [38,68,70,71]).

Although within-family GWASs were not genetically confounded when the populations were separate (because cis- and trans-LDs were equal, as discussed above), they become genetically confounded in the admixed population, as all trans-LD is eliminated by random mating in the admixed population, leaving an excess of cis-LD relative to trans-LD that biases effect-size estimates (Eqs (6) and (7)). As in the case of the population GWAS, these biases will be zero on average if allele frequency differences between the ancestral populations were due to drift. However, after admixture, they will still inflate the average squared effect-size estimate (and thus the variance of effect-size estimates), which will thereafter decline in subsequent generations as the cis-LD is gradually broken down by recombination (Eq (17); Fig 5).

In comparing the average squared effect-size estimate in a population and a family-based GWAS, we observe that the value in the population GWAS rapidly declines to approximately the same level as the value in the within-family GWAS, despite the former having started at a much higher level in the initial admixed population (Fig 5). The explanation is that LD between unlinked loci confounds effect-size estimation in the population GWAS but not the within-family GWAS, such that (i) the average squared effect-size estimate from the population GWAS is initially much higher than that from a within-family GWAS, because it is inflated by LD between many more pairs of loci; and (ii) the average squared effect-size estimate from the population GWAS declines more rapidly, because LD between unlinked loci is broken down more rapidly than LD between linked loci.

**Allele frequency divergence due to selection or phenotype-biased migration.**   In addition to drift, and as discussed above, selection and phenotype-biased migration can generate systematic, signed (effect-size correlated) LD, which would lead to systematic cis-LD in the descendent admixed population. These would lead to larger inflations of genetic variance and genetic correlations than would be expected had allele frequency divergence between the ancestral populations been due to drift alone, and would complicate interpretations of genetic correlations as being due to pleiotropy. Moreoever, if the admixed population is more than a few generations old such that LD between unlinked loci but not linked loci has largely been broken down, then population- and family-based estimates of these quantities might be similar.

**Spurious genetic correlations due to confounding in population-based PGSs.**   Factors other than selection and phenotype-biased migration can also generate non-pleiotropic genetic correlation signals in family-based studies of admixed populations. In fact, the use of confounded population GWAS effect sizes can be sufficient. As an example of the confounding of genetic correlations in admixed populations due to a confounded GWAS for one trait, consider the GIANT-GWAS height PGS. Owing to confounding within Europe [20,21], the height PGS showed large differences between Northern Europeans and sets of individuals sampled in other locations, such as the African 1,000 genomes samples [72]. This confounding generated

a spurious, systematic correlation between height effect sizes and allele frequency differences across populations, with height-increasing alleles that are more common among Northern Europeans being assigned larger effects [20]. As a result, in a PGS constructed from these effect-size estimates, larger PGS values are predictive of greater North European ancestry. Now imagine a sibling-based study performed in a sample with recently admixed "European" and "non-European" ancestry—African Americans, for example. An individual with a larger value than their sibling for the GIANT height PGS will, on average, carry more "European" ancestry. In African Americans, there will also be a systematic association of lighter skin pigmentation with recent "European" ancestry, and selection on skin pigmentation will have driven a signed difference in allele frequencies between European and West African ancestors. Putting these observations together, the GIANT height PGS, being predictive of the degree of European ancestry, may well be predictive of skin pigmentation differences between African American sibling pairs (Eq (11)), leading to the naive and incorrect conclusion that height and skin color are causally linked. In reality, this result would reflect the fact that alleles predicted to increase height and alleles that affect skin color are in systematic effect-signed admixture LD, as in Eq (16), as a consequence of stratification-biased effect-size estimates from the GIANT European GWAS.

## 3.4 Stabilizing selection

Stabilizing selection—selection against deviations from an optimal phenotypic value—is thought to be common [73], and has recently been argued to be consistent with the genetic architectures of many human traits [60]. By disfavoring individuals with too many or too few trait-increasing alleles, stabilizing selection generates negative LD among alleles with the same directional effect on the trait [10]. Thus, stabilizing selection will attenuate GWAS effect-size estimates at genotyped loci that tag these causal loci.

To quantify these biases, we consider the model of Bulmer [10,11], in which a large number of loci contribute to variation in a trait under stabilizing selection, with the population having adapted such that the mean trait value is equal to the optimum. Under this model, stabilizing selection rapidly reduces variance in the trait by generating, within each generation, negative cis- and trans-LD among trait-increasing alleles; this negative LD is then partially transmitted (because of recombination) to the next generation in the form of cis-LD.

If we make the simplifying assumption that all loci have equal effect sizes, then the equilibrium reduction in trait variance in a given generation, $-d^*$ (where $d^*<0$), measured before the action of selection in that generation, can be calculated as a function of the genic variance $V_g$, the environmental noise $V_E$, the strength of stabilizing selection $V_S/V_P$ (scaled according to the phenotypic variance $V_P$), and the harmonic mean recombination rate, $\bar{c}_h$, among loci underlying variation in the trait ([11]; S1 Text Section S3.4). Selection within the same generation will reduce the trait variance further, to a degree that can also be calculated as a function of $V_g$, $V_E$, $V_S/V_P$, and $\bar{c}_h$ (S1 Text Section S3.4; [11]).

Under the assumption of equal effect sizes across loci, we calculate in S1 Text Section S3.4 the average per-locus attenuation bias in effect-size estimates induced by stabilizing selection, $(\alpha_l - \hat{\alpha}_l)/\alpha_l$. In a population GWAS performed before selection has acted in the sampled individuals' generation, the attenuation bias is approximately

$$\frac{\alpha_l - \hat{\alpha}_l^{\mathrm{pop}}}{\alpha_l} = -\frac{d^*}{V_g},$$

while in a population GWAS performed after selection has acted in the sampled individuals'

generation, the attenuation bias is approximately

$$\frac{\alpha_l - \hat{\alpha}_l^{\text{pop}}}{\alpha_l} = -\frac{d^*(1 + 2\bar{c}_h)}{V_g};$$

that is, a factor $1 + 2\bar{c}_h$ greater than in the pre-selection population GWAS.

In a within-family GWAS, the average proportionate bias in effect-size estimation is approximately

$$\frac{\alpha_l - \hat{\alpha}_l^{\text{fam}}}{\alpha_l} = -\frac{d^*(1 - 2\bar{c}_h)}{V_g},$$

irrespective of whether the GWAS is performed before or after selection has acted (or is ongoing) in the generation from which the sample is drawn. The attenuation bias in a family-based GWAS is therefore smaller than that in a pre-selection population GWAS by a factor of $1 - 2\bar{c}_h$, and smaller than that in a post-selection population GWAS by a factor of $(1 - 2\bar{c}_h)/(1 + 2\bar{c}_h)$.

Thus, the bias in effect-size estimation can be calculated given estimates of the phenotypic variance and heritability of the trait, the harmonic mean recombination rate, and the strength of stabilizing selection (S1 Text Section S3.4). In the Methods, making some simplifying assumptions about the genetic architecture of the trait in question, we calculate an approximate value $\bar{c}_h \approx 0.464$ for humans. Using this value, Fig 6 shows the average proportionate reduction in GWAS effect-size estimates for various strengths of stabilizing selection and heritabilities of the trait. The range of selection strengths was chosen to match that inferred for human traits by Sanjak and colleagues [74].

Attenuation of effect-size estimates is larger if stabilizing selection is stronger or if the trait is more heritable. Taking height as an example, heritability is ~0.8, $V_P \approx 7 \text{cm}^2$, and Sanjak and colleagues [74] estimate a sex-averaged strength of stabilizing selection of $V_S/V_P \approx 30$. From these values, we calculate that a pre-selection population GWAS would systematically underestimate effect sizes at loci that causally influence height by about 2.7% on average, in the absence of other sources of LD (Fig 6A), while a post-selection population GWAS would systematically underestimate effect sizes by about 5.2%, on average (Fig 6B). More generally, within the range of reasonable strengths of stabilizing selection inferred by Sanjak and colleagues [74], we calculate average attenutations of pre-selection population-based effect-size estimates of up to 5%

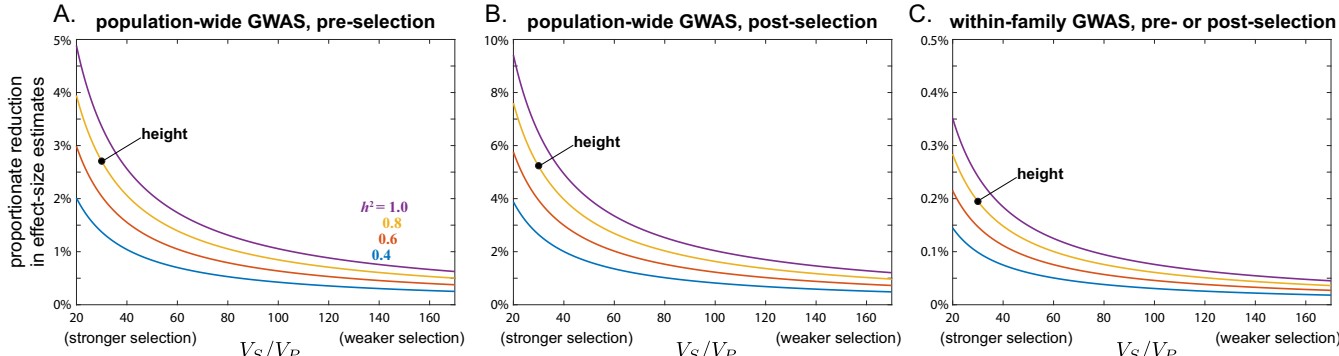

**Fig 6. Stabilizing selection attenutates GWAS effect-size estimates.** The calculations displayed here assume that genetic variation in the trait is contributed by 1,000 loci of equal effect spaced evenly along the human genome. Stabilizing selection is stronger if the width of the selection function scaled by the phenotypic variance, $V_S/V_P$, is smaller. The placement of the point for human height assumes a heritability of 0.8 and a strength of stabilizing selection of $V_S/V_P = 30$, as estimated in ref. [74]. Details of these calculations can be found in S1 Text Section S3.4. Note the different scales of the y-axes in (A), (B), and (C).

for highly heritable traits ($h^2 \approx 1$) under strong stabilizing selection ($V_S/V_P \approx 20$), down to 0.25% for less heritable traits ($h^2 \approx 0.4$) under weak stabilizing selection ($V_S/V_P \approx 170$) (Fig 6A); the analogous range is 0.5% to 10% for post-selection population-based effect-size estimates (Fig 6B).

Given the estimate $\bar{c}_h \sim 0.464$, the proportionate bias that stabilizing selection induces in within-family GWASs is expected to be a fraction $1 - 2\bar{c}_h \approx 7.2\%$ that in pre-selection population GWASs and $(1 - 2\bar{c}_h)/(1 + 2\bar{c}_h) \approx 3.7\%$ that in post-selection population GWASs. Thus, for height, a within-family GWAS would underestimate effect sizes by only about 0.2% on average (Fig 6C).

The quantitative importance of these biases will vary by application. In situations where the goal is gene discovery, for example, 5% to 10% reductions in effect-size estimates are unlikely to flip the statistical significance of variants with large effects on a trait. However, the attenuations in effect-size estimates caused by stabilizing selection are systematic across loci, and therefore could substantially affect aggregate quantities based on these estimates. For example, the range of average reductions in population effect-size estimates calculated above for human traits would translate to reductions in naive estimates of SNP-based heritabilities of between 0.5% and 20% (approximately 5.4% in the case of height using effect-size estimates from a pre-selection population GWAS; approximately 10% using estimates from a post-selection population GWAS). If effect sizes are estimated by within-family GWAS, on the other hand, the reductions in these SNP-based heritability estimates would be much smaller.

As a further example, by generating negative LD between alleles with the same directional effect on the trait, the impact of stabilizing selection opposes, and therefore masks, the genetic impact of assortative mating [75]. A practical consequence is that stabilizing selection will tend to attenuate estimates of the strength of assortative mating based on GWAS effect sizes, which often use cross-chromosome correlations of PGSs (e.g., [57,76]). In humans, the phenotypic correlation among mates for height has been measured at about approximately 0.25 [58]. In S1 Text Section S3.4, we calculate that estimates of this correlation based on cross-chromosome correlations in PGSs will be biased downwards by about 20% (to approximately 0.20) because of stabilizing selection on height, if selection has not yet acted in the measured generation, and by about 40% (to 0.15) if selection has acted in the measured generation. Were assortative mating weaker, or stabilizing selection stronger, the genetic impact of assortative mating would be masked to an even greater extent (S1 Text Section S3.4).

As in our analysis of assortative mating above, if stabilizing selection ceases in some generation, the negative LD that built up during the period of stabilizing selection will decay over subsequent generations, rapidly for pairs of loci on different chromosomes and more slowly for linked pairs of loci. Patterns of selection on human traits have changed over time—for example, the strength of stabilizing selection on birth weight has relaxed [77]. In general, therefore, patterns of confounding reflect a composite of contemporary and historic processes.

## 3.5 Sibling indirect effects

Indirect effects of siblings' genotypes on each other's phenotypes are known to be a potential source of bias in sibling-based GWASs [34,52], and can be measured and corrected for only if, in addition to sibling genotypes, parental genotypes are also available, either directly or via imputation [27,34]. To generate intuition for their impact on GWASs, we consider a simple model of indirect sibling effects in the absence of G×E interactions and other confounding effects, focusing on a single-locus model for simplicity. We suppose that the indirect effect of an individual's phenotype on their sibling's (same) phenotype is $\beta$, so that the phenotypes of 2

siblings $i$ and $j$ can be written as follows:

$$Y_i = Y^* + \alpha g_i + \beta Y_j + \epsilon_i,$$

$$Y_j = Y^* + \alpha g_j + \beta Y_i + \epsilon_j. \tag{18}$$

Taking their difference and rearranging, we find that

$$\Delta Y = \frac{\alpha}{1+\beta}\Delta g + \frac{1}{1+\beta}\Delta \epsilon. \tag{19}$$

Therefore, in the absence of genetic confounding and G×E interactions, a sibling-based association study would return an effect-size estimate of

$$\hat{\alpha}^{\text{sib}} = \frac{\alpha}{1+\beta} \tag{20}$$

in expectation. Thus, if sibling indirect effects are synergistic ($\beta>0$), they lead to underestimation of the direct genetic effect at the locus. In contrast, if sibling indirect effects are antagonistic ($\beta<0$), they lead to overestimation of the direct genetic effect.

How would a population GWAS be affected by the same sibling indirect effects? Sibling $i$'s phenotype can be written as follows:

$$Y_i = Y^* + \alpha g_i + \beta Y_j + \epsilon_i$$

$$= Y^* + \alpha g_i + \beta(Y^* + \alpha g_j + \beta Y_i + \epsilon_j) + \epsilon_i$$

$$\Rightarrow Y_i = \frac{1}{1-\beta^2}\left(Y^{**} + \alpha g_i + \alpha\beta g_j + \epsilon_i + \beta\epsilon_j\right), \tag{21}$$

where $Y^{**} = (1+\beta)Y^*$. Therefore, if we were to randomly choose 1 sibling from each sibship and estimate the effect size at the locus using a population association study across families, we would obtain

$$\hat{\alpha}^{\text{pop}} = \frac{\text{Cov}(g_i, Y_i)}{\text{Var}(g_i)} = \frac{1}{1-\beta^2}\left(\alpha + \alpha\beta r_g^{\text{sibs}}\right), \tag{22}$$

where $r_g^{\text{sibs}} = \text{Cov}(g_i, g_j)/\text{Var}(g_i)$ is the genotypic correlation between sibs at the locus. Sibling indirect effects alter the effect-size estimate in a population GWAS via 2 channels. The first is through the factor $1/(1-\beta^2)$ in Eq (22), which reflects second-order feedbacks of an individual's phenotype on itself, via the sibling (see also ref. [78]). Since $1/(1-\beta^2)>1$, these feedbacks act to exacerbate the effects of causal alleles. For example, if sibling indirect effects are antagonistic ($\beta<0$), then a sibling with a large trait value will tend to indirectly reduce the trait value of their sibling, which in turn will indirectly further increase the trait value of the focal individual. This channel therefore pushes population GWASs towards overestimating the magnitude of direct genetic effects.

The other channel by which sibling indirect effects can influence a population GWAS is driven by the genotypic correlation among siblings, and is easiest to understand if we assume that sibling indirect effects are weak ($\beta^2 \ll 1$). In this case, $\hat{\alpha}^{\text{pop}} \approx \alpha + \alpha\beta r_g^{\text{sibs}}$. Since the genotypic correlation $r_g^{\text{sibs}} > 0$, this channel of sibling indirect effects has the opposite effect to the one it has on a sibling GWAS: if sibling indirect effects are synergistic ($\beta>0$), the population GWAS overestimates the direct genetic effect at the locus, while if sibling indirect effects are

antagonistic ($\beta < 0$), the population GWAS underestimates the direct genetic effect. The reason for this difference is that a sibling GWAS is based on siblings whose genotypes differ at the focal locus, and whose genotypic values are therefore anticorrelated. If sibling indirect effects are synergistic ($\beta > 0$), they will tend to attenuate the phenotypic differences between such siblings, and therefore attenuate effect-size estimates. In contrast, because siblings' genotypes are positively correlated across the entire population, synergistic sibling indirect effects ($\beta > 0$) will tend to exacerbate phenotypic differences across families, leading a population GWAS to overestimate effect sizes.

## 4 Discussion

It has long been recognized that population GWASs in humans can be biased by environmental and genetic confounding [1,4]. Currently, population GWASs attempt to control for these confounds by focusing on sets of individuals that are genetically more similar and by controlling for population stratification. However, these controls are imperfect and are not always well defined. For example, controlling for genome-wide patterns of population stratification based on common alleles does not control for the genetic and environmental confounding of rare variants [79]. Work on genetic confounding has uncovered increasing evidence that assortative mating may be leading to large biases in estimates of direct genetic effects and to large genetic correlations for a number of traits [22,37,57]; moreover, it can often be unclear whether genetic signals of assortative mating are due to trait-based mate choice or some other more general form of genetic confounding (e.g., [80]). Additionally, while we have focused primarily on genetic confounding, for a number of traits there are also signals of residual environmental confounding in GWAS signals [31,32,35,81]. Thus, subtle and often interwoven forms of genetic and environmental confounding remain a major issue in many GWASs [34], compromising the interpretation of GWAS effect-size estimates and downstream quantities such as SNP heritabilities and genetic correlations.

Effect-size estimates from within-family GWASs are less affected by these various confounds. They are not subject to environmental confounding across families, because the environments of family members are effectively randomized with respect to within-family genetic transmission. As we have shown, family-based estimates should also suffer substantially less from genetic confounding, because genetic transmission at unlinked loci (but not linked loci) is randomized by independent assortment of chromosomes in meiosis. Nonetheless, family-based GWASs can suffer from residual genetic confounding as well as sibling indirect effects (in sibling-based designs), and causal interpretations of the estimates they produce are complicated by G×E and G×G interactions [44]; they also raise a number of further conceptual problems that we discuss below.

### Sources of genetic confounding

Genetic confounding is caused by long-range LD between loci that affect the trait or traits under study. To illustrate the potential for genetic confounds to bias GWAS effect-size estimates, we have considered several sources of long-range LD. Some of these—assortative mating, selection on GWAS traits, and phenotype-biased migration—can cause systematic directional biases in GWAS effect-size estimates. Others, such as neutral population structure, cause random biases that influence the variance of effect-size estimates and related quantities. Assortative mating and neutral population structure have received considerable theoretical attention in the GWAS literature (e.g., [22,37,38,57]). Here, we have further outlined how both selection and phenotyped-biased migration can drive systematic genetic confounding that

may not be well accounted for by current methods of controlling for stratification (see also ref. [81]).

We wish to emphasize stabilizing selection in particular as a potential source of systematic confounding in GWASs. Stabilizing selection has been well studied in the quantitative genetics literature but less so in the context of GWASs, despite its expected ubiquity. By selecting for compensating combinations of trait-increasing and trait-decreasing alleles, stabilizing selection generates negative LD between alleles with the same directional effect on the trait [10,11], and can therefore bias GWAS effect-size estimates downwards. While the potential for stabilizing selection to confound effect-size estimation has been noted (e.g., [66,75,82]), the resulting biases have not, to our knowledge, been quantified. Our calculations suggest that these downward biases could, for some human traits, be as large as 5% to 10% systematically across all causal loci in population GWASs. While biases of this magnitude are unlikely to compromise some goals of GWASs, such as gene discovery, they could be quantitatively problematic for other GWAS aims, such as estimation of SNP heritabilities and the strength of assortative mating. Moreover, while our results pertain to (a particular model of) stabilizing selection, many kinds of selection generate LD between genetically distant loci—in fact, only multiplicative selection among loci does not ([83], pgs. 50 and 177). Therefore, the general result that selection can generate genetic confounding will hold more broadly. Even in the absence of natural and sexual selection, the ascertainment of samples for a population- or family-based GWAS is a form of selection. When participation in a GWAS sample is based partly on a genetically influenced phenotype [84], this sample-selection bias will generate LD between loci underlying participation. In family studies, this participation bias can violate the assumption of random assignment of genotypes, and thus potentially undermine the interpretation of these studies' results.

For a given genotyped locus in a GWAS, there is no bright line between local "tagged" LD and long-range confounding LD, and one reasonable objection to the approach taken here is that that we have used an arbitrary definition of the causal loci that are locally tagged by a genotyped locus ($L_{local}$ in Eq (2)). All of the sources of genetic confounding that we have considered generate LD among causal loci both within and across chromosomes. Under these models, the within-chromosome LD that is generated is, in a sense, a continuation of the LD generated across chromsomes (moving from a recombination rate = 0.5 to $\leq 0.5$). Thus, while investigators may prefer some looser definition of "local" when thinking about genotyped GWAS loci as tag SNPs, to extend that definition to include all loci on the same chromosome as the SNP would, by reasonable interpretation, be to include confounding into the desired estimator.

The extent to which the absorption of genetic confounding in estimated effect sizes is a problem depends on the application. In the case of polygenic prediction, the absorption of environmental effects, indirect effects, and the effects of untyped loci throughout the genome can help to improve prediction accuracy, although this does come at a cost to interpretability (and potentially also to portability across contexts). For GWAS applications focused on understanding genetic causes and mechanisms, the biases in effect-size estimates and spurious signals of pleiotropy among traits generated by genetic confounding will be more problematic.

## Indirect genetic effects

Family GWASs are often interpreted as providing the opportunity to ask to what extent parental genotypes (or other family genotypes) causally affect a child's phenotype ("genetic nurture" [27]). Viewed in this way, the association between untransmitted parental alleles and the child's phenotype would seem, at first, a natural estimate of indirect genetic effects.

In practice, however, if the population GWAS suffers from genetic and environmental confounds, then the estimated effects of untransmitted alleles will absorb that confounding in much the same way that estimates of direct genetic effects from a population GWAS do (Eqs (8) and (9)) [50]. For example, in the case of assortative mating, a given untransmitted allele is correlated with alleles that were transmitted both by this parent and by their mate, and these transmitted alleles can directly affect the offspring's phenotype. Thus, while family-based estimates of direct genetic effects benefit from the randomization of meiosis and from controlling for the environment, family-based estimates of indirect genetic effects lack both of these features and should be interpreted with caution. Indeed, recent work using parental siblings to control for grandparental genotypes has shown that little of the estimated "indirect genetic effect" may be causally situated in parents [36]. With empirical estimates of indirect genetic effects potentially absorbing a broad set of confounds [34,85], and few current studies of indirect effects having designs that allow such confounding to be disentangled, it is premature—and potentially invalid—to interpret associations of untransmitted alleles causally in terms of indirect genetic effects [3]. Rather, they should be treated agnostically in terms of "non-direct" effects.

## Direct genetic effects

Mendelian segregation provides a natural randomization experiment within families [86], and so crosses in experimental organisms and family designs have long been an indispensable tool to geneticists in exploring genetic effects and causation. Growing concerns about GWAS confounding and the increasing availability of genotyped family members have led to a return of family-based studies to the association study toolkit [2]. Family-based estimates of direct genetic effects are often interpreted as being unbiased and discussed in terms of the counterfactual effect of experimentally substituting one allele for another [34,87,88].

As we have shown, family-based GWASs are indeed less subject to confounding than population-based GWASs: in the presence of genetic and environmental confounding, the family-based estimate of the effect size at a given locus provides a much closer approximation to the true effects of tightly linked causal loci than a population-based estimate does. The family-based estimate is not biased by environmental variation across families and avoids the correlated effects of the many causal loci that lie on other chromosomes. Still, the family-based estimate does absorb the effects of non-local causal loci on the same chromosome, and so cannot truly be said to be free of genetic confounding. Rather than considering a single allele being substituted between individuals, a better experimental analogy for the effect-size estimate would be to say that we are measuring the mean effect of transmission of a large chunk of chromosome surrounding the focal locus, potentially carrying many causal loci.

In addition, while within-family GWASs offer these advantages, in other ways, they move us further away from the questions about the sources and causes of variation among unrelated individuals that motivate population GWASs in the first place. Indeed, the presence of confounding and of G×E/G×G interactions introduces a number of conceptual issues in moving from within-family GWAS to the interpretation of differences among individuals from different families [44,89,90]. For example, in the presence of genetic confounding, the estimated effect of a causal allele of interest will depend on a set of weights: its LD to many other causal alleles. Family-based approaches weight these LD terms differently to population-based approaches, which, we argue, can complicate the interpretation of these estimates. For example, when previously isolated populations admix, same-ancestry alleles will be held together in long genomic blocks until these are broken up by recombination, which will happen very quickly for alleles on different chromosomes but more slowly for alleles on the same

chromosome. A few generations after admixture, therefore, cross-chromosome ancestry LD will largely have dissipated, but contiguous ancestry tracts will still span substantial portions of chromosome lengths. Since both population and within-family GWASs are similarly confounded by the same-chromosome LD, their mean squared effect sizes will be similar in this case (Fig 5). Bearing in mind that the LD resulting from admixture is not present in the source populations, it becomes unclear which weighting of ancestry LD is appropriate if we want to interpret the resulting effect-size estimates as direct effects. As this example illustrates, while family-based GWASs are a useful device for dealing with confounding, it is not always obvious how to interpret the quantities that they measure.

A number of additional complications arise when, to compensate for the small effect sizes of individual loci, researchers combine many SNPs into a PGS and study the effects of PGSs within families (or use them as instruments in mendelian randomization analyses). For one, SNPs are usually chosen for inclusion in the PGS on the basis of their statistical significance in a population GWAS. This approach prioritizes SNPs whose effect-size estimates are amplified (or even wholly generated) by confounding (for an example of how this leads to residual environmental confounding in applications of sibling-based effect-size estimates, see ref. [79]). Second, the weights given to SNPs that are included in the PGS absorb the effects of confounding, and this confounding is heterogeneous across SNPs. Thus, when we study the correlates of trait-A PGS differences between siblings in the presence of GWAS confounding, we are not observing the average phenotypic outcomes of varying the genetic component of trait A between siblings. Rather, we are varying a potentially strangely weighted set of genetic correlates of trait A.

An observation that a population-GWAS PGS is predictive of phenotypic differences among siblings demonstrates that the PGS SNPs tag nearby causal loci, but beyond that, interpretation is difficult. Notably, if there is cross-trait assortative mating for traits A and B, but no pleiotropic link between the traits, then some of the SNPs identified as significant in a GWAS on trait A may be tightly linked to loci that causally affect trait B but not trait A. If these loci are included in the trait-A PGS, then when we study the effect of variation in the trait-A PGS on sibling differences, we are accidentally absorbing some components of the variation in trait B across siblings. Thus, we might observe a correlation between the trait-A PGS and differences in trait B between siblings, and this correlation may be lower than is observed at the population level, without there existing any pleiotropic (or causal) link between A and B. These effects can be exacerbated if the 2 traits have different genetic architectures (Fig 4). Instead of using a set of SNPs and weights from a population GWAS, genetic correlations between traits due to pleiotropy could be estimated from the correlation of effect sizes estimated within families [33]. Given current sample size constraints in family-based studies, the confidence intervals on these estimates are large. Moreover, significant family-based correlations need not reflect pure pleiotropy, since, as we have shown, they are not completely free of genetic confounding due to intra-chromosomal LD.

Also complicating the interpretation of family-based effect-size estimates are various types of interactions. Indirect effects between siblings can bias family estimates of direct genetic effects (Eq (20); [2,34,52]) in ways that are conceptually different from the biases they introduce to population-based estimates (Eq (22)). These sibling effects can potentially be addressed with fuller family information (e.g., parental genotypes in addition to sibling genotypes [27,34]).

In summary, family-based studies are a clear step forward towards quantifying genetic effects, with large-scale family studies carrying the potential to resolve long-standing issues in human genetics. However, these designs come with their own sets of caveats, which will be important to understand and acknowledge as family-based genetic studies become a key tool in the exploration of causal effects across disparate fields of study.

## 5 Methods

All simulations were carried out in SLiM 4.0 [53]. Code is available at https://doi.org/10.5281/zenodo.10520811.

For the purpose of carrying out sibling association studies in our simulations, we assumed a simple, monogamous mating structure: each generation, each female, and each male is involved in a single mating pair, and each mating pair produces exactly 2 offspring (who are therefore full siblings). To maintain the precisely even sex ratio required by this scheme, we assumed that a quarter of mating pairs produce 2 daughters, a quarter produce 2 sons, and half produce a son and a daughter. Population sizes were chosen to ensure that these numbers of mating pairs were whole numbers, and mating pairs were permuted randomly each generation before assigning brood sex ratios (to ensure that no artifact was introduced by SLiM's indexing of individuals).

Each generation, per-locus effect size estimates were calculated for both population-wide and sibling GWASs. The former were calculated as the regression of trait values on per-locus genotypes, while the latter were calculated as the regression of sibling differences in trait values on sibling differences in per-locus genotypes.

In all simulations, the total population size was $N = 40{,}000$.

### Assortative mating

For our general cross-trait assortative mating setup, traits 1 and 2 are influenced by variation at sets of bi-allelic loci $L_1$ and $L_2$, respectively. The effect sizes of the reference allele at locus $l$ on traits 1 and 2 are $\alpha_l$ and $\beta_l$, respectively. An individual's PGS is then $P_1 = \sum_{l \in L_1} g_l \alpha_l$ for trait 1 and $P_2 = \sum_{l \in L_2} g_l \beta_l$ for trait 2. In all the scenarios we simulated, traits had heritability 1, so that individuals' trait values are the same as their PGSs.

Our aim is to simulate a scenario where assortative mating is based on females' values for trait 1 and males' values for trait 2, such that, across mating pairs, the correlation of the mother's PGS for trait 1, $P_1^m$, and the father's PGS for trait 2, $P_2^f$, is a constant value $\rho$ (in all of our simulations, $\rho = 0.2$). To achieve this, we use an algorithm suggested by Zaitlen and colleagues [69]: At the outset, we choose an accuracy tolerance $\varepsilon$ such that, if by some assignment of mates the correlation of their PGSs falls within $\varepsilon$ of the target value $\rho$, we accept that assigment. Each generation in which assortative mating occurs, we rank females in order of their PGSs for trait 1, and males in order of their PGSs for trait 2. We then calculate the PGS correlation across mating pairs, $\rho_0$, if females and males were matched according to this ranking. If this (maximal) correlation is smaller than the upper bound of our target window ($\rho_0 < \rho + \varepsilon$, which very seldom occurred in our simulations), then females and males mate precisely according to their PGS rankings and we move on to the next generation. If, instead, $\rho_0$ exceeds $\rho + \varepsilon$, then we follow the following iterative procedure until we have found a mating structure under which the correlation of PGSs falls within $\varepsilon$ of the target value $\rho$.

First, we choose initial "perturbation sizes" $\xi_0$ and $\xi_1 = 2\xi_0$. Suppose that, in iteration $k$ of the procedure, the perturbation size is $\xi_k$ and the chosen mating structure leads to a correlation among mates of $\rho_k$. If $|\rho_k - \rho| < \varepsilon$, we accept the mating structure and move on to the next generation. Otherwise, we choose a new perturbation size $\xi_{k+1}$: (i) if $\rho_{k-1}, \rho_k > \rho$, then $\xi_{k+1} = 2\xi_k$; (ii) if $\rho_{k-1} > \rho > \rho_k$ or $\rho_{k-1} < \rho < \rho_k$, then $\xi_{k+1} = (\xi_{k-1} + \xi_k)/2$; (iii) if $\rho_{k-1}, \rho_k < \rho$, then $\xi_{k+1} = \xi_k/2$. Once we have chosen $\xi_{k+1}$, for each individual we perturb their PGS (trait 1 for females; trait 2 for males) by a value chosen from a normal distribution with mean 0 and standard deviation $\xi_{k+1}$, independently across individuals. We then rank females and males according to their perturbed PGSs, and calculate the correlation $\rho_{k+1}$ of their true PGSs if they mate according to

this ranking. (Since, in our experience, there can be substantial variance in the $\rho_{k+1}$ values that result from this procedure, we repeat it 5 times and choose the mating structure that produces the value of $\rho_{k+1}$ closest to the target value $\rho$.) We then decide if another iteration—i.e., another perturbation size $\xi_{k+2}$—is required.

## Fig 2. Cross-trait assortative mating for traits with the same genetic architecture

In the simulations displayed in Fig 2, $\rho$ = 0.2, and traits 1 and 2 have identical but non-overlapping genetic architectures: $L_1$ and $L_2$ are distinct sets of 500 loci each, with $\alpha_l$ = 1 and $\beta_l$ = 0 for $l \in L_1$, and $\alpha_l$ = 0 and $\beta_l$ = 1 for $l \in L_2$. Loci in $L_1$ and $L_2$ alternate in an even spacing along the physical (bp) genome. Fig 2A shows results for the "single chromosome" case where the recombination fraction between adjacent loci is $c$ = 1/999 in both sexes (such that the single-chromosome genome receives, on average, one crossover per transmission). Fig 2B shows results for the case where recombination fractions between loci are calculated from the human female and male linkage maps generated by Kong and colleagues [54]. In both cases, we assumed no crossover interference.

At each locus, the initial frequency of the reference allele was 1/2, with reference alleles assigned randomly across diploid individuals and independently across loci such that, in expectation, Hardy–Weinberg and linkage equilibrium initially prevail. The assortative mating algorithm above was run for 19 generations, with a target correlation $\rho$ = 0.2, a tolerance parameter $\varepsilon$ = $\rho$/100, and an initial perturbation size $\xi_0 = 4[\max(\{\{P_1^m\}, \{P_2^f\}\}) - \min(\{\{P_1^m\}, \{P_2^f\}\})]$. Thereafter, assortative mating was switched off, with mating pairs (still monogamous) being chosen randomly.

## Fig 3. Same-trait assortative mating

The algorithm we followed to ensure assortative mating of a given strength was the same as that for Fig 2 above, but here traits 1 and 2 are identical, and 1,000 loci underlie variation in the trait, and are evenly spread along the physical genome. The effect size of the reference allele at each locus was drawn from a normal distribution with mean 0 and standard deviation 1, independently across loci. The initial frequency of the reference allele at each locus was drawn, independently across loci, from a uniform distribution on [$MAF$, 1−$MAF$]; in our simulations, we chose a minimum minor allele frequency of $MAF$ = 0.1. Since here we are interested in quantifying the mean squared effect size estimate, which is directionally affected by drift-based local LD that may not be present in our initial configuration, we allowed 150 generations of random mating before switching on assortative mating (only the final 20 generations of this random mating burn-in are displayed in Fig 3). Assortative mating occurred for 20 generations, after which random mating occurred for a further 20 generations.

## Fig 4. Cross-trait assortative mating for traits with different architectures

For Fig 4A, we again followed a similar procedure to that for Fig 2 above, but now, while traits 1 and 2 have distinct genetic bases, the numbers of loci contributing variation to traits 1 and 2 are $|L_1|$ = 100 and $|L_2|$ = 1,000. Trait-1 loci are placed evenly along the physical genome, with trait-2 loci then evenly spaced among the trait-1 loci; we used the human linkage map for these simulations. At both trait-1 and trait-2 loci, the initial frequency of the focal allele was drawn from a uniform distribution on [$MAF$, 1−$MAF$], with $MAF$ = 0.1. At trait-2 loci, true effect sizes were randomly drawn from a normal distribution with mean zero and standard deviation 1; at trait-1 loci, true effect sizes were randomly drawn from a normal distribution with mean

zero and standard deviation $\sqrt{10}$, so that traits 1 and 2 have equal genic variances. After a burn-in of 150 generation of random mating, assortative mating was switched on. We performed a population GWAS at the end of the period of random mating and after 20 generations of assortative mating. These GWASs were performed across 1,000 replicate trials, with the effect size estimates then pooled across trials. From these, we estimated the densities of the absolute values of effect size estimates using Matlab's kernel density estimator ksdensity, specifying that the support of the distributions be positive. For Fig 4B, as a control, we also carried out the same simulations with $|L_1| = |L_2| = 1,000$, drawing effect sizes from a standard normal distribution.

## Fig 5. Population structure and admixture

We wished first to simulate a situation where 2 populations of size $N/2$ have been separated for a length of time such that the value of $F_{ST}$ between them is some predefined level (in our case, a mean $F_{ST}$ per locus of 0.1). To do so without having to run the full population dynamics of 2 allopatric populations for a prohibitively large number of generations, we simply assigned allele frequencies to achieve the desired level of $F_{ST}$. We assumed 1,000 loci spread evenly over the physical genome. At each locus $l$, we chose an "ancestral" frequency $p_l^a$ for the reference allele independently from a uniform distribution on $[MAF, 1-MAF]$, with $MAF = 0.2$. We then perturbed this allele frequency in populations 1 and 2 by independent draws from a normal distribution with mean 0 and variance $2p_l^a(1 - p_l^a)F_{ST}$; if a perturbed allele frequency fell below 0 or above 1, we set it to 0 or 1, respectively. The population dynamics described above, with monogamous mating pairs chosen randomly, were then run for 50 generations.

In generation 50, the 2 populations merge, forming an admixed population of size $N$. The same population dynamics, with monogamous mating pairs chosen randomly, were then run for a further 50 generations.

## Fig 6. Stabilizing selection

To calculate the bias in GWAS effect size estimation caused by stabilizing selection, we must first calculate the harmonic mean recombination rate. We focus on humans and consider only the autosomal genome. The set of loci underlying variation in the trait is $L$, which we apportion among the 22 autosomes according to their physical (bp) lengths (as reported in GRCh38.p11 of the human reference genome; https://www.ncbi.nlm.nih.gov/grc/human/data?asm=GRCh38.p11). For each chromosome, we spread its allotment of loci evenly over its sex-averaged genetic (cM) length, using the male and female linkage maps produced by Kong and colleagues [54]. (We use genetic lengths instead of physical lengths because, were we to spread loci evenly over the physical lengths of the chromosomes, some pairs of adjacent loci on some chromosomes might have a sex-averaged recombination fraction of 0, in which case the harmonic mean recombination rate would be undefined.) For each pair of linked loci, the recombination rate between them was estimated separately from the male and female genetic distance between them using Kosambi's map function [91]. Pairs of loci on separate chromosomes have a recombination fraction of 1/2. With the sex-averaged recombination fraction $c_{ll'}$ thus calculated for every pair of loci $(l,l')$, the harmonic mean recombination fraction was calculated as $\bar{c}_h = \binom{|L|}{2} / \left( \sum_{l,l'} \frac{1}{c_{ll'}} \right)$, where $\binom{|L|}{2} = |L|(|L| - 1)/2$ is the number of pairs of distinct loci in $L$.

Performing this calculation with $|L| = 1,000$ loci, we obtain an estimate of $\bar{c}_h \approx 0.464$ for human autosomes. Substituting this estimate into S1 Text Eqs (S.106), (S.107), and (S.108) then defines the curves plotted in Figs 6A, 6B and 6C, respectively.

## Supporting information

**S1 Text. Supplementary information.**
(PDF)

**S1 Fig. Cross-trait assortative mating influences effect-size estimates at loci that affect the study trait, although this influence is second-order relative to that on effect-size estimates at loci that do not affect the study trait but do affect the other trait involved in assortative mating (cf. Fig 2; note the scales of the y-axes).** Simulations are the same as in Fig 2; the code can be found at https://doi.org/10.5281/zenodo.10520811.
(PDF)

## Acknowledgments

We thank Jeremy Berg, Doc Edge, Arbel Harpak, Gibran Hemani, Hanbin Lee, Molly Przeworski, Alex Young, and members of the Coop lab for helpful discussions and comments on earlier drafts.

## Author Contributions

**Conceptualization:** Carl Veller, Graham M. Coop.

**Formal analysis:** Carl Veller, Graham M. Coop.

**Writing – original draft:** Carl Veller, Graham M. Coop.

**Writing – review & editing:** Carl Veller, Graham M. Coop.

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
