## [Editor Report · Decision Letter 0]

31 Mar 2023

Dear Dr Veller, 

Thank you for submitting your manuscript entitled "Interpreting population and family-based genome-wide association studies in the presence of confounding" for consideration as a Research Article by PLOS Biology.

Your manuscript has now been evaluated by the PLOS Biology editorial staff, as well as by an academic editor with relevant expertise, and I'm writing to let you know that we would like to send your submission out for external peer review.

Once your full submission is complete, your paper will undergo a series of checks in preparation for peer review. After your manuscript has passed the checks it will be sent out for review. To provide the metadata for your submission, please Login to Editorial Manager (https://www.editorialmanager.com/pbiology) within two working days, i.e. by Apr 04 2023 11:59PM.

Kind regards,

Roli Roberts

Roland Roberts, PhD

Senior Editor

PLOS Biology

rroberts@plos.org

---

## [Decision Letter · Decision Letter 1]

12 Jun 2023

Dear Dr Veller,

Thank you for your patience while your manuscript "Interpreting population and family-based genome-wide association studies in the presence of confounding" was peer-reviewed at PLOS Biology. It has now been evaluated by the PLOS Biology editors, an Academic Editor with relevant expertise, and by three independent reviewers. 

In light of the reviews, which you will find at the end of this email, we would like to invite you to revise the work to thoroughly address the reviewers' reports.

You’ll see that reviewer #1 is positive, but raises two main issues; one is about the methodologies that you consider for family-based GWAS, and the other is your definition of bias/confounding; he suggests remedies for these. He also thinks you should shorten the manuscript and describe the simulations better. Reviewer #2 is very positive and only has a couple of points, one of which overlaps with rev #1 (the definition of bias). Reviewer #3 is also broadly positive, but thinks that you need to tie it in with underlying mechanistic processes (he also takes issue with the term “genetic confounding”). He has a number of other points that mostly relate to real-world examples and implications that might make the paper more accessible.

Given the extent of revision needed, we cannot make a decision about publication until we have seen the revised manuscript and your response to the reviewers' comments. Your revised manuscript is likely to be sent for further evaluation by all or a subset of the reviewers.

**IMPORTANT - SUBMITTING YOUR REVISION**

*Re-submission Checklist*

*Published Peer Review*

*PLOS Data Policy*

*Blot and Gel Data Policy*

Sincerely,

Roli Roberts

Roland Roberts, PhD

Senior Editor

PLOS Biology

rroberts@plos.org

REVIEWERS' COMMENTS:

Reviewer #1:

[identifies himself as Alexander Young]

IMPORTANT: See also the comments on the attached PDF!

The authors present a manuscript that evaluates the influence of a variety of factors on both standard and family-based GWAS methods. The manuscript represents an important contribution to the literature on understanding what different GWAS designs are estimating, a topic of considerable importance since so much in human genetics relies upon the output of these studies. The results that I found most enlightening were those detailing how different population genetic processes can lead to forms of within-chromosome linkage disequilibrium (LD) that influence family-based and standard GWAS designs differently. Also important is the authors' warning about the interpretation of within-family analyses of polygenic predictors (PGS) derived from standard GWAS owing the influence of confounding in the original GWAS summary statistics. 

While I found the manuscript made many valuable points, there are some issues with the theoretical results on family-based GWAS and on the definitions of 'bias' and 'confounding' the authors use. I outline the major issues here, with detailed comments given in an annotated PDF. 

The first issue is the authors' derivation of results for family-based GWAS using transmitted and non-transmitted alleles. The authors define this method as proceeding by performing two separate univariate regressions on transmitted and untransmitted parental alleles, with direct effects estimated from the difference of these regression coefficients (A1.4). This is not the methodology used in family-based GWAS studies (e.g. Young et al. 2022) or in studies of indirect genetic effects (e.g. Kong et al. 2018). The method used in Young et al. 2022 is a regression of offspring phenotype jointly onto offspring genotype (or PGS) and combined parental genotype (or PGS). This is equivalent to a joint regression onto genotypes computed from the transmitted and non-transmitted parental alleles, with the direct effect computed by subtracting the coefficient on the non-transmitted genotype from the coefficient on the transmitted genotype, the method used by Kong et al. 2018. The equivalence between these two models is shown in Section 7.2 of the Supplementary Information of Okbay et al. 2022. While this is equivalent to performing separate regressions on transmitted and non-transmitted genotypes under random-mating, which means transmitted and non-transmitted genotypes (or PGS) are uncorrelated, this is not true in general, including for many of scenarios considered in the manuscript. I'm not sure if this makes much difference in some scenarios (e.g. analysis of a single variant affecting a polygenic phenotype affected by assortative mating) but could make a substantial difference under others (e.g. strong population structure, or a PGS that is strongly correlated between parents). However, most of the results in the main text use the sibling design, for which the author's derivation is correct as far as I can tell.

The second issue is how the authors define bias or confounding. This is a tricky issue in the case of GWAS designs since they are estimating marginal effects, which are inherently defined in terms of linkage disequilibrium with other variants. The authors define the true effect to be that of the local region around the focal SNP, i.e. the region with recombination fraction approximately 0. But this is not what GWAS is estimating even under random-mating with direct genetic effects only: it is estimating a linear combination of direct genetic effects of causal variants on the same chromosome, with weight in proportion to the LD between the focal variant and the causal variant. It would be better to define the true target parameter as such (as would be obtained from standard GWAS under random-mating with no indirect genetic effects) and then to examine bias in relation to that parameter. 

A related issue is the authors examine the impact of various population genetic processes on a quantity they term the "heterozygosity weighted average squared effect size estimate". An issue with this quantity is that GWAS gives a biased estimate of the true value of this even under random-mating and direct genetic effects only (e.g. Figure 3). It therefore makes it hard to assess the impact of various population genetic processes on the bias introduced into this term, and it makes me doubt whether it is even a sensible thing to estimate with marginal GWAS effect estimates, whether from standard or family-based GWAS. It would both be clearer and increase the impact of this paper on the field if the authors could measure bias in some quantity or method that is generally used and (at least thought to be) understood, such as SNP heritability as estimated by LDSC. LDSC attempts to 'undo' the fact that marginal GWAS effect estimates are the outcome of LD with variants that are physically close to the focal variant when estimating SNP heritability. Results demonstrating how various population genetic processes impact LDSC estimates of SNP heritability (which should be approximately unbiased under a model of random mating and direct genetic effects only) from standard and family-based GWAS would be far more interesting and interpretable than results on 'heterozygosity weighted average squared effect size estimate'. (I appreciate there is a complicating factor that the true heritability changes with these different population genetic processes, but that is something I'm sure the authors could find a sensible way of dealing with and communicating. True SNP heritability could, for example, be computed from a PGS computed from the true direct effects of the causal SNPs.)

I also think the authors are stretching the definition of 'confounding' when they claim GxE and GxG interactions lead to confounding in family-based GWAS estimates. Under interaction models, all marginal effects are some sort of average effect, with the average taken over some distribution of the interacting factors. Such marginal effects are not typically thought of as 'confounded' by interactions in statistical modeling. The authors point that family-based and standard GWAS are estimating different average effects under certain interaction models is valid, but I don't think this can be construed as 'confounding' in family-based GWAS. 

The authors should also include a description of the simulations in the manuscript. I followed the link to the git repository, and did not find it immediately enlightening. 

Finally, the main text is rather long. While I thought the authors generally did a good job of making a technical topic more comprehensible, the writing could be tightened up significantly. 

Alexander Young. 

Reviewer #2:

This is a fantastic study by Veller & Coop that lays out how population structure impacts estimates of SNP effects and other derived quantities obtained across various experimental designs (population and within-family designs).

I only have minor comments.

My first comment relates to the word "bias", which seems overly used in this manuscript (x47 times). I will illustrate my point by simply talking about heritability. The heritability of a trait has a different meaning (and expectation) depending on whether the population undergoes random mating or not. Therefore, assuming that the random mating heritability is the true heritability and the other one biased does not seem to be the appropriate language. Even more so, given that a randomly mating population is a mere abstraction that would probably never exist in nature. The paper has a lot of this fallacious dichotomy. This is a minor comment so the only action that I request here is for the authors to emphasize that the notion of bias is subjective and will depend on what the application is.

My second comment regards the effectiveness of PCA to correct stratification. The authors echo what was reported before that PCA adjustment may not fully remove confounding. However, this is only half true as the power of PCA depends on sample size. Interestingly, Price (2006) clearly made that point and gave a rule of thumb to appreciate when PCA starts detecting (sub)structures. However, that argument tends to get lost in the literature. My suggestion is to emphasize that the power of PC depends on sample size or, equivalently, to add that PCA is inefficient in (meta-analyses of) small samples.

"Transmitted genotype" - Only alleles are transmitted. Please fix.

Reviewer #3:

[identifies himself as Gibran Hemani]

Veller and Coop provide a very detailed and careful theoretical analysis of subtle biasing structures in different GWAS designs, which I personally found to be valuable and educational. The authors use their analysis to propose caution over family-based methods for genetic association analysis, however a lack of contextualising with empirically known parameters of biasing mechanisms made this feel difficult to easily accept or action.

Main comments

Mechanistic processes. I did struggle with explanations such as "Genetic confounding is caused by long-range LD between loci that affect the trait or traits under study" which are used throughout - it equates an unobservable state of the data with an observable state of the data, however it would be better to equate a mechanistic process (e.g. stabilising selection, cross-trait assortative mating etc) with an observable state of the data (long range LD). Perhaps avoid the term genetic confounding. It appears that there are cross-trait mechanisms that introduce long-range LD, and within-trait mechanisms that drive issues at the causal locus. So start with the mechanisms, list what types of LD they can generate, and summarise the extent of bias likely arising from those mechanisms.

Empirical support. Many papers have focused on showing bias in population-based estimates. However few examine bias in family-based estimates, though increasing numbers of family-based estimates are appearing. Can you pinpoint specific examples of family-based estimates which are particularly liable to be incorrect due to some of these biasing mechanisms?

Sensitivity analysis. A useful outcome from this would be to allow researchers to examine the sensitivity of an estimate by being able to plug in parameters into some of the equations to explore how much e.g. assortative mating would be required to give rise to the observed results from their study.

Polygenicity. The simulations used quite low polygenicity (100 or 1000 causal variants). The more socially driven traits that are liable to many of the biasing mechanisms are likely much more polygenic. How sensitive are some of the conclusions to greater polygenicity.

Tagging SNPs vs other forms of LD. I think it's quite well understood that sentinel variants detected by genome wide scans are not necessarily the causal variants and that they may slightly underestimate the true causal effect of the locus. The other forms of LD bias are much less well known in my opinion. In the exposition it would be anchoring to have a clear distinction between these two ideas at the outset.

Minor comments

line 31 - it's more accurate to describe assortative mating as a collider/selection bias rather than confounding.

line 52 - could say that indirect/dynastic effects are particularly difficult to control using population-level covariates

line 52 - 'there is no bright line separating dynastic, environmental and genetic confounding' - The mechanism of dynastic effects is quite specific, and forms a special case of both environmental confounding. I think it would serve the reader better to clarify these distinctions rather than imply that they can't be clarified.

line 118 - perhaps italicise "other parent" 

Trans-LD - can you provide a more clear mechanistic explanation about how this manifests? It seems like cross-trait assortative mating is the main driver? In any case I would suggest renaming some of these terms. 

Line 249 - is it just sampling error that would drive differences in coupling and repulsion heterozygote frequencies or some other mechanism? If sampling error, does it contribute towards asymptotic bias? If there's a mechanism that drives differences then it would be more direct to re-factor this equation in terms of parameters for that mechanistic process.

Line 333 - Determining if a slope is non-zero depends on hypothesis testing which has a power component. There hasn't been any discussion of the magnitude of the biasing terms or their standard errors, i.e. for what sample size of sibling pairs, and for what level of cross trait assortative mating, polygenicity etc would be required in order for a sibling pair PGS slope to be determined to be non-zero? 

Line 394 - the h^2 rho term is a bit unclear when there are two traits in cross-trait AM, each potentially having different h^2

Figure 3 - I think squared effects are being used to proxy h2-like metrics, however there is an issue with bias in h2 due to changes in phenotypic variance due to AM. Kemper et al 2021 discuss how there isn't an obvious 'true' value for h2

---

## [Decision Letter · Decision Letter 2]

12 Jan 2024

Dear Dr Veller,

Thank you for your patience while we considered your revised manuscript "Interpreting population- and family-based genome-wide association studies in the presence of confounding" for publication as a Research Article at PLOS Biology. This revised version of your manuscript has been evaluated by the PLOS Biology editors, the Academic Editor, and one of the original reviewers.

Based on the review, we are likely to accept this manuscript for publication, provided you satisfactorily address the remaining points raised by the reviewer. Please also make sure to address the following data and other policy-related requests:

IMPORTANT:

a) Please attend to the remaining comments from reviewer #1.

b) This is a minor point, but when reading your Title, several of us stumbled over the use of the gerund "confounding" at the end. It feels like an adjective with a missing noun, rather than a noun in its own right. Would "confounds" or "confounders" or some such be easier for a casual reader to parse?

c) Many thanks for providing the simulation code in Github. However, because Github depositions can be readily changed or deleted, please make a permanent DOI’d copy (e.g. in Zenodo) and provide this URL (see below).

d) Please cite the location of the simulation code clearly in all relevant main and supplementary Figure legends (Figs 2, 3, 4, 5, S1, I think), e.g. “The simulation code used to generate this Figure can be found in https://doi.org/10.5281/zenodo.XXXXX”

e) I note that you mention two of the reviewers (Young, Hemani) in the Acknowledgements. While we appreciate the sentiment, this is against PLOS policy, so please could you remove this?

We expect to receive your revised manuscript within two weeks. 

*Published Peer Review History*

*Press*

Sincerely,

Roli Roberts

Roland Roberts, PhD

Senior Editor,

rroberts@plos.org,

PLOS Biology

CODE POLICY

Per journal policy, as the code that you have generated is important to support the conclusions of your manuscript, we require that you make it available without restrictions upon publication. Please ensure that the code is sufficiently well documented and reusable, and that your Data Statement in the Editorial Manager submission system accurately describes where your code can be found.

DATA NOT SHOWN?

REVIEWER'S COMMENTS:

Reviewer #1:

[identifies himself as Alexander Young]

The authors have done a good job at justifying their choice of the heterozygosity weighted squared effect size as their metric of potential bias. While I personally would still be interested to see simulation results applying LDSC to the different scenarios examined in this manuscript, that is perhaps beyond the scope of what is already a comprehensive manuscript and which now includes a theoretical treatment of potential LDSC biases within the authors' theoretical framework. 

The authors have corrected their model for trio-based GWAS estimation. However, I was looking at their equations for the non-transmitted coefficient, and I'm not sure that it is totally correct. Looking at Equation 9, the 'genetic confounding direct' component appears to imply that this coefficient can pickup some of the direct effect of the focal variant. If I set \\lambda=l, so that c_{\\lambda,l}=0, this equation implies a contribution from a term equal to (\\tilde{D}'_{l,l}+\\tilde{D}_{l,l})\\alpha_l^d, i.e. the sum of trans-LD in the offspring and parent generations at locus l times the direct genetic effect of the locus. I do not see how the non-trasnmitted coefficient can pick up part of the direct effect of the focal variant in a joint regression of phenotype onto genotype and parental genotype, since the direct effect of the focal locus will be fully captured by the offspring genotype at that locus, even when there is trans-LD within the locus. Can the authors comment on this? 

(There is a simpler route to deriving the trio-GWAS regression coefficients that avoids having to deal with the complexities of the full joint regression calculations. One can reformulate the regression in terms of the offspring deviation genotype — g_l-(g_l^m+g_l^f)/2 — and the sum of the parents' genotypes. Since these terms have zero covariance even under non-random mating, one can obtain the coefficients for the univariate regressions of phenotype onto these two terms and then take the sum and rearrange in order to obtain the results for the joint regression of phenotype onto offspring and parental genotype. I take this approach in some lectures I gave available here: https://www.youtube.com/watch?v=Imc_gJyhahQ&ab_channel=SocialScienceGeneticsAssociationConsortium)

Also, on lines 1456-1458, the authors invoke the Frisch-Waugh-Lovell theorem to justify performing a regression on the offspring deviation genotype in order to derive the 'direct genetic effect' estimate. But they state that they only need to residualise the offspring genotype for the midparent genotype, whereas the FWL theorem also requires that the phenotype be residualised with respect to the midparent genotype. 

A couple of minor things:

line 1553: should this be a '+' not an '='? 

Line 2350: 'teh' -> 'the'

---

## [Editor Report · Decision Letter 3]

19 Jan 2024

Dear Dr Veller,

Thank you for the submission of your revised Research Article "Interpreting population- and family-based genome-wide association studies in the presence of confounding" for publication in PLOS Biology. On behalf of my colleagues and the Academic Editor, Priya Moorjani, I'm pleased to say that we can in principle accept your manuscript for publication, provided you address any remaining formatting and reporting issues. These will be detailed in an email you should receive within 2-3 business days from our colleagues in the journal operations team; no action is required from you until then. Please note that we will not be able to formally accept your manuscript and schedule it for publication until you have completed any requested changes.

Sincerely,

Roli Roberts

Senior Editor

PLOS Biology

rroberts@plos.org